# Functional divergence of a bacterial enzyme promotes healthy or acneic skin

Irshad A. Hajam[1,11], Madhusudhanarao Katiki[2,11], Randall McNally[2,3,11], María Lázaro-Díez [1,4,11], Stacey Kolar[2,5,11], Avradip Chatterjee [2], Cesia Gonzalez[1], Mousumi Paulchakrabarti[6], Biswa Choudhury[6], J. R. Caldera [1,2,7], Trieu Desmond[1,8], Chih-Ming Tsai[1], Xin Du [1], Huiying Li[9], Ramachandran Murali [2] ✉ & George Y. Liu [1,10] ✉

Acne is a dermatologic disease with a strong pathologic association with human commensal *Cutibacterium acnes*. Conspicuously, certain *C. acnes* phylotypes are associated with acne, whereas others are associated with healthy skin. Here we investigate if the evolution of a *C. acnes* enzyme contributes to health or acne. Two hyaluronidase variants exclusively expressed by *C. acnes* strains, HylA and HylB, demonstrate remarkable clinical correlation with acne or health. We show that HylA is strongly pro-inflammatory, and HylB is modestly anti-inflammatory in a murine (female) acne model. Structural and phylogenic studies suggest that the enzymes evolved from a common hyaluronidase that acquired distinct enzymatic activity. Health-associated HylB degrades hyaluronic acid (HA) exclusively to HA disaccharides leading to reduced inflammation, whereas HylA generates large-sized HA fragments that drive robust TLR2-dependent pathology. Replacing an amino acid, Serine to Glycine near the HylA catalytic site enhances the enzymatic activity of HylA and produces an HA degradation pattern intermediate to HylA and HylB. Selective targeting of HylA using peptide vaccine or inhibitors alleviates acne pathology. We suggest that the functional divergence of HylA and HylB is a major driving force behind *C. acnes* health- and acne- phenotype and propose targeting of HylA as an approach for acne therapy.

Acne vulgaris affects four of five individuals sometime during their lifetime. Predisposition to acne is dependent on both host and environmental factors. Among these, the contribution of skin commensal *C. acnes* has been debated since both healthy and acne-prone subjects are robustly colonized with *C. acnes*. Recent skin microbiome characterizations of *C. acnes* strain populations, from acne patients and individuals with healthy skin, demonstrate the importance of *C. acnes* genetic elements as a major acne determinant[1–3], as development of

[1]Department of Pediatrics, University of California San Diego, San Diego, CA 92093, USA. [2]Department of Biomedical Sciences, Research Division of Immunology, Cedars-Sinai Medical Center, Los Angeles, CA 90048, USA. [3]Vault Pharma Inc., 570 Westwood Plaza, Los Angeles, CA 90025, USA. [4]AIDS Research Institute (IrsiCaixa). VIRus Immune Escape and VACcine Design (VIRIEVAC) Universitary Hospital German Trias i Pujol Crta Canyet s/n 08916, Badalona, Barcelona, Spain. [5]Pharmacology at Armata Pharmaceuticals, Inc., Marina del Rey, CA 90292, USA. [6]GlycoAnalytics Core, University of California San Diego, San Diego, CA 92093, USA. [7]Department of Pathology & Laboratory Medicine, UCLA Health & David Geffen School of Medicine, Los Angeles, CA 90095, USA. [8]School of Pharmacy, University of California San Francisco, San Francisco, CA 94143, USA. [9]Department of Molecular and Medical Pharmacology, Crump Institute for Molecular Imaging, David Geffen School of Medicine, UCLA, Los Angeles, CA 90095, USA. [10]Division of Infectious Diseases, Rady Children's Hospital, San Diego, CA 92123, USA. [11]These authors contributed equally: Irshad A. Hajam, Madhusudhanarao Katiki, Randall McNally, María Lázaro-Díez, Stacey Kolar. ✉e-mail: ramachandran.murali@csmc.edu; gyliu@health.ucsd.edu

acne is strongly associated with certain *C. acnes* strains and phylotypes. Accordingly, *C. acnes* strains have been categorized based on their health or acne association. Consistent with this classification, studies of select health- and acne- associated *C. acnes* strains have demonstrated greater induction of inflammation or potentially deleterious responses by acne-associated strains compared to health-associated strains in vitro[4,5].

Subsequent metagenomics studies unveiled sets of genes that are prominently present in acne- or health-associated strains of *C. acnes*[1,6], thereby ushering in a new front in the quest for the understanding of acne pathogenesis. Yet, acne pathogenesis is poorly understood, hampered by the absence of a robust animal model and poor survival of *C. acnes* in rodents. We recently addressed growth of *C. acnes* in mice by applying human synthetic sebum to murine skin with infection that permitted persistence of *C. acnes*[7]. We further demonstrated remarkable and uniformly enhanced immunopathogenicity of acne-associated strains compared to health-associated strains in the model. Using this model, we set to address the question – what genetic element(s) drove the divergence of *C. acnes* disease or health association?

Among the candidate factors revealed by comparative genomic studies of health- versus acne-associated strains, we were particularly intrigued by a matrix-degrading enzyme hyaluronidase (Hyl). In mammals, Hyl generates HA fragments that mediate inflammation via TLR2/4 signaling[8], which is a major proinflammatory pathway in acne pathogenesis[9]. *C. acnes* encodes two variants of the Hyl enzyme, HylA and HylB, which are selectively expressed by acne- and health-associated strains, respectively[10]. Herein, we sought to gain a greater understanding of their relationship and their contribution to *C. acnes* health and acne association. We showed that HylA hydrolyze HA into large-sized HA fragments that drive robust TLR2-dependent inflammatory pathology. In contrast, HylB degrades hyaluronic acid (HA) exclusively to HA disaccharides leading to reduced acne immunopathology. Structural and phylogenic studies suggest that the enzymes evolved from a common hyaluronidase that acquired diverging enzymatic properties. We showed that selective inhibition of HylA by peptide inhibitor or by vaccination alleviates acne pathology, thus pointing to a virulence-based approach to acne treatment.

## Results

### *C. acnes* hyaluronidases contribute to healthy or acneic skin

To assess the potential importance of HylA or HylB enzyme in clinical acne, we surveyed all *hylA* and *hylB* genes in the NCBI and profiled their association with health- and acne-associated *C. acnes* strains (Supplementary Table 1). *C. acnes* strains are classified into phylotypes that have different predilections for association with acne[6,11,12]. Based on whole genome sequences[6,13], phylotype clades IA-1, IA-2, IB-1, IB-2 and IC are associated with acne, whereas clade II is closely associated with healthy skin (Fig. 1a). Skin microbiome analysis based on 16S rRNA sequencing demonstrates a strong association of ribotype (RT) 2/6 with health and RT4/5 with acne disease (Supplementary Table 1)[6]. As shown in Fig. 1a, *hylA* gene is found in acne-associated clades and *hylB* in health-associated clades, supporting their potential to contribute to acne or health.

To directly query the role of *hylA* and *hylB* in acne, we generated in-frame allelic exchange of *hylB* and *hylA* in health-associated (HL110PA3, Clade II, RT6) and acne-associated (HL043PA1, Clade IA-2, RT5) strains, respectively. From our prior study, these two strains represent the least and most acnegenic strains of the panel of RT2/6 and RT4/5 health and acne strains, respectively, tested in our acne mouse model[7]. We verified *Hyl* gene deletion by sequencing and loss of Hyl activity on HA plate assay (Supplementary Fig. 1). We applied both WT and mutant strains to our murine acne model, and surveyed disease score and tissue cytokines production (Fig. 1b–f and Supplementary Fig. 2a–g). Independent of bacterial burden (Fig. 1b), the *hylA* deletion mutant induced dramatic reduction in disease score and

proinflammatory cytokines compared to the parent acne-associated strain (Fig. 1c–f). Conversely, the *hylB* deletion mutant demonstrated a modest increase in disease score and pro-inflammatory cytokines compared to the health-associated parental strain (Fig. 1c–f), consistent with the interpretation that HylB has anti-inflammatory properties. Notably, when we compared the immunopathology of acne induced by the prototype acne- and health-associated *C. acnes* strains, the difference between strains was abrogated or modestly reversed in the absence of Hyl enzymes, pointing to the important contribution of the Hyl variants to phenotypic differences between the health and acne strains.

We provided corroboration of HylA proinflammatory phenotype by complementation of Δ*hylA* with WT HylA recombinant protein (rHylA) injection (Fig. 1g–j and Supplementary Fig. 2g). Overall, our findings suggest that the two Hyl variants play a major role in promoting or mitigating of acne immunopathology with HylA demonstrating a dominant proinflammatory role and HylB conferring a modest anti-inflammatory effect.

### HylA and HylB enzymes have distinct HA degradation pattern and efficiency

We next asked how HylA and HylB developed such distinct inflammatory properties. Reports have shown that mammalian and *Streptomyces hyalurolyticus* Hyl enzymes produce HA fragments larger than 4-mers that contribute to the induction of pro-inflammatory cytokines[14,15]. Recently, we reported that bacterial pathogens (Group B *Streptococcus, S. pneumoniae* and *S. aureus*) generate Hyls that degrade proinflammatory HA strictly to non- or anti-inflammatory disaccharides (HA-2)[16]. Hence, Hyls could mediate different inflammatory outcomes depending on the size of HA degradation products they generate.

We incubated supernatant from HL110PA3 or HL043PA1, or recombinant HylA or HylB with high molecular weight (HMW) HA for 1 or 24 hr and analyzed the resulting HA products using chromatography (Supplementary Fig. 3a–g and Supplementary Fig. 4a–f). HylB enzymatic activity rapidly produced predominantly HA-2 (Fig. 2a, Supplementary Fig. 3 and Supplementary Fig. 5). Tetrasaccharide HA (HA-4) was briefly observed at 5 min (Supplementary Fig. 3b), but thereafter, only HA-2 was detected as the degradation reaction continued.

In comparison, HylA produced different-sized oligosaccharides throughout the reaction course, including HA−4 and hexasaccharide HA (HA−6) and HA-2 (Fig. 2b). Notably, at 24 hr, HA-4, H-6 and higher MW HA persisted (Supplementary Fig. 4c). We performed exhaustive digestion with HylA-containing supernatant for up to 6 days and found that digestion was still incomplete. Although lower amounts of HylA compared to HylB were secreted into supernatants (Supplementary Fig. 6), the differences in HA breakdown products were not overcome by increased (recombinant) enzyme concentration (Supplementary Fig. 3e). These differences in the degradation rate and HA fragmentation pattern between HylA and HylB are consistent with findings from another group on three HylA (phylotype IA) and two HylB (phylotype IB and II) producing strains[10]. These observations suggest there are fundamental differences in degradation mechanisms between these two enzymes.

### HylA and HylB divergence promotes distinct mechanisms of HA degradation

Unlike human Hyls, enzymes secreted by commensal or pathogenic bacteria reported to date, degrade HA strictly to HA-2[16]. However, a single cluster of bacteria stands out in the Hyl phylogenetic tree that can generate fragments larger than disaccharides. This cluster includes environmental bacteria, such as *Streptomyces* that degrade HA into large fragments (Supplementary Fig. 7). Cutibacterium (Propionibacterium), both a human commensal and an environmental

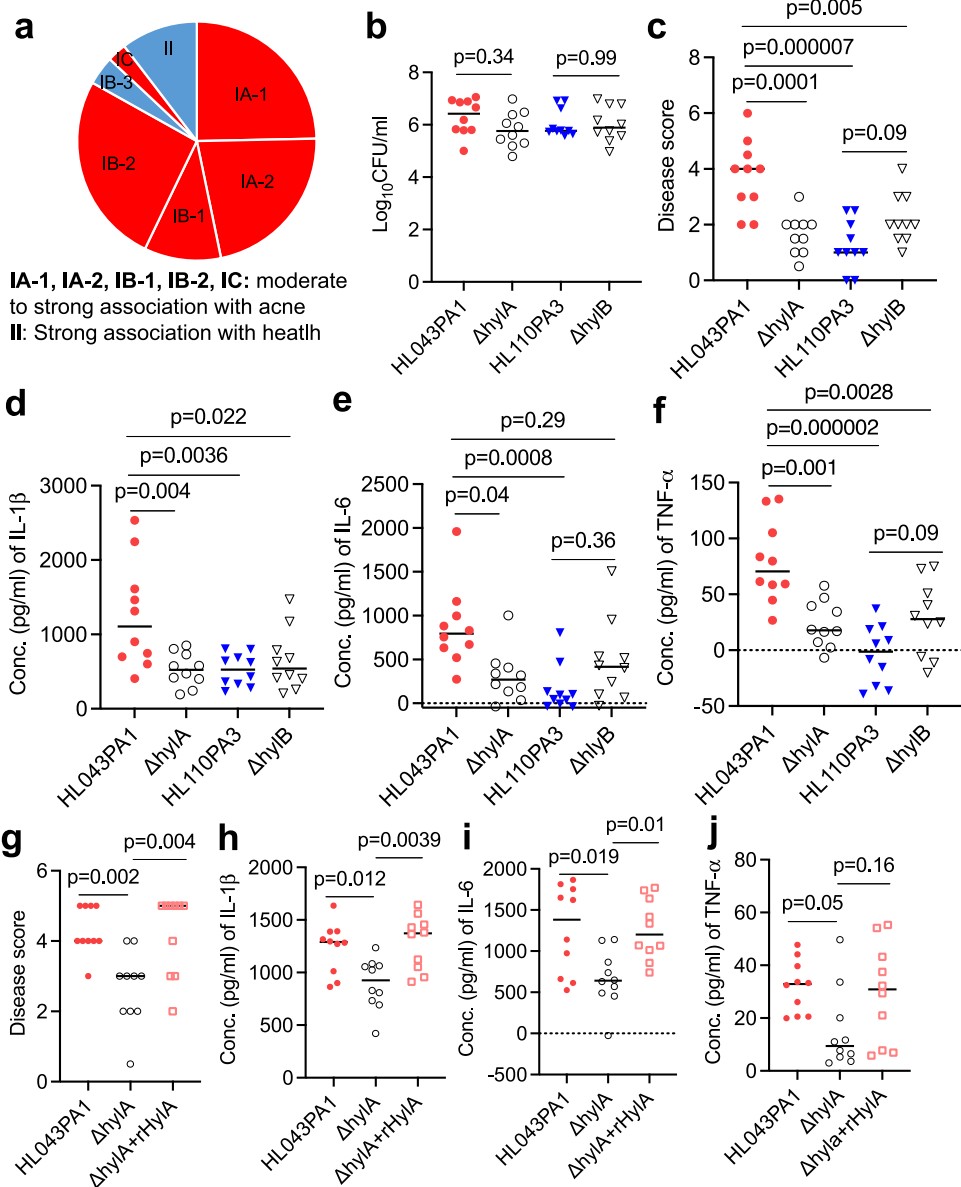

**Fig. 1 | HylA enzyme is a major virulence factor in acne pathogenesis. a** pie chart showing health-and-acne-associated *C. acnes* phylotypes and association with *hylA* or *hylB gene*[6,13]. **b–f** CD1 mice (*n* = 10) were infected intradermally (i.d.) with 2x10⁷CFU WT (HL043PA1 or HL110PA3) or isogenic mutant (Δ*hylA* or Δ*hylB*) *C. acnes*, followed by topical application of sebum daily. Bacterial burden (**b**), disease score (**c**), and cytokines (**d**–**f**) at 2 d (48 h) post-infection. **g–j** CD1 mice (*n* = 10) were infected as above with either HL043PA1, Δ*hylA* or Δ*hylA* plus recombinant (r) HylA

protein (10 μg). Disease score (**g**), and tissue cytokines (**h**, **j**) at 2d post-infection. **b–j** Data were from two independent experiments with each data point representing one mouse. Bars denote median. The data in **b**, **c** and **e**–**i** were analyzed by one-way ANOVA with Tukey's post-hoc test. The data in **d** and **j** were analyzed by non-parametric Kruskal-Wallis one-way ANOVA test. Source data are provided as a Source Data file.

bacterium, clusters alongside *Streptomyces*. This association raises the question if proinflammatory HylA derives from *Streptomyces* and anti-inflammatory HylB from bacterial pathogens such as *Streptococci*. HylA and HylB are 74% identical in nucleotide sequence and 74% identical in amino acid sequence (Supplementary Fig. 8). Comparison of HylA and HylB to homologous enzymes across related species show that the two enzymes most likely originated within Cutibacterium lineage (Supplementary Fig. 7).

### Structures of HylA and HylB reveal high structural similarity between them and with homologous glycosaminoglycan-degrading lyases from other bacteria

To understand the structural basis for the differences in the hyaluronate lyase activities of HylA and HylB, we solved the X-ray crystal structures of HylA Y285F and wild-type HylB to 2.05 Å and 2.1 Å, respectively (Supplementary Table 2 and Fig. 2c). HylA Y285F is a catalytically deficient form of the enzyme, and the structure is hereafter referred to as "HylA." Befitting enzymes with 74% identity between them, the structures are highly similar, overlaying with a r.m.s.d. of 0.8 Å over 751 residues (both HylA molecules in the crystallographic asymmetric unit vs. HylB). Typical of hyaluronate lyases, HylA and HylB consist of a mostly α-helical N-terminal domain, a C-terminal domain comprising mainly of β-strands, and a catalytic site in a large cleft predominantly within the N-domain (Fig. 2c, d). The catalytic sites overlay closely, containing many elements conserved in hyaluronate lyases, including two conserved tryptophans (HylA/B Trp161/157 and Trp162/158), several positively charged residues, and the three residues of the catalytic triad (Asn226/222, His276/272, and

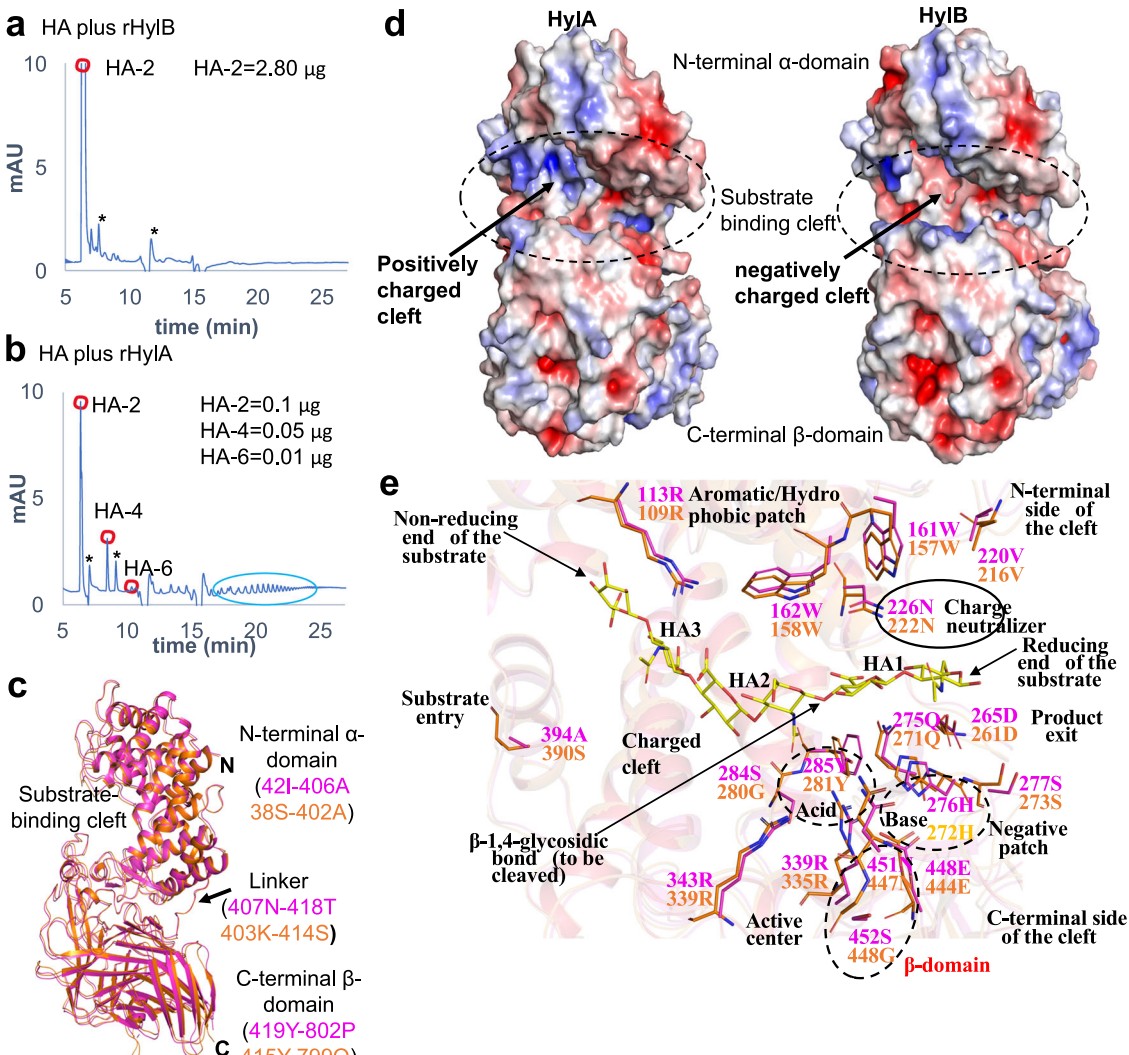

**Fig. 2 | HA degradation and structural features of HylA and HylB enzymes.**
**a, b** HPLC profile of HMW-HA (2 mg/ml) digested for 24 hr with rHylB or rHylA (0.35 ug). Digested HA peaks (HA-2, −4 and −6) were quantified using known concentrations of purified HA oligosaccharides (see Supplementary Figs. 3, 4). Larger-sized HA fragments, highlighted with a green circle, were visualized only with rHylA-digested HA. Asterisk (*) in **a, b** represents non-specific peaks. The results are representative of at least 2 independent experiments. **c** comparison of HylA (PDB: 8FYG) and HylB (PDB: 8FNX) HylB crystal structures. HylA and HylB are shown by cartoon in magenta and orange, respectively. The structural domains, linker and substrate-binding cleft are labeled. **d** the electrostatic surface view is shown for the HylA and HylB crystal structures. The substrate binding cleft is highlighted in dashed oval. Red and blue correspond to potentials of −5 kT $e^{-1}$ and 5 kT $e^{-1}$, respectively. The electrostatic potentials were calculated by APBS in PyMol. **e** the residue-wise similarities and differences at the substrate-binding cleft of HylA and HylB. The HA-6 ligand is taken from the *Streptococcus pneumoniae* Hyl (SpHyl) crystal structure (PDB: 1LOH). Source data are provided as a Source Data file.

Tyr285/281[17,18] (Fig. 2e). We further solved the structure of the catalytically deficient Y281F mutant of HylB to a resolution of 2.1 Å (Supplementary Table 2); this structure was solved in a different space group than wild-type HylB (P1 with two molecules in the crystallographic asymmetric unit vs. P4₁2₁2 and one molecule for wild-type). Though crystallized with different crystal-packing interactions, HylB Y281F is nearly identical in conformation to wild-type HylB (r.m.s.d 0.6 Å, wild-type vs. both Y281F molecules) (Supplementary Fig. 9a–e). We were unable to produce crystals of HylA or HylB complexed with HA fragments; indeed, the three structures reported here display open catalytic clefts likely incompatible with binding the substrate (Fig. 3).

The crystal structures show that *C. acnes* HylA and HylB share high structural similarity with glycosaminoglycan lyases from gram-positive bacteria, including hyaluronate lyases from *Streptococcus agalactiae* and *Streptococcus pneumoniae*, xanthan lyases from *Bacillus* sp. strain GL1 and *Paenibacillus nanensis*, and chondroitin AC lyases from *Streptomyces coelicolor* and *Arthrobacter aurescens*[17–28] (Fig. 3a–c). Each

of these enzymes is homologous to HylA and HylB, with sequence identities ranging from 23−37% vs. HylA, and are structurally homologous as well, with root mean square deviations vs. HylA ranging from 2.2−3.4 Å. Additionally, the geometry of catalytic residues conserved within these enzymes is maintained in HylA and HylB (Fig. 3d, e).

To obtain further insight into the functional divergence of HylA and HylB, we identified four residue positions in the catalytic cleft that differ between HylA and HylB. Two of these residue pairs (HylA Arg397/HylB Val393 and HylA Ser116/HylB Glu112) are located deep in the cleft in proximity to the β−D-glucuronic acid moiety at the non-reducing end of the putative bound HA, and contribute to the binding cleft of HylA displaying a more positively-charged surface than that of HylB (Fig. 2d). The other two pairs (HylA Asp345/HylB Asn341 and HylA Glu346/HylB Gly342) lie closer to the predicted position of the preceding β−D-glucuronic acid moiety.

Next, we examined these residue pairs by mutating the HylA residues to match their HylB counterparts and vice versa, and

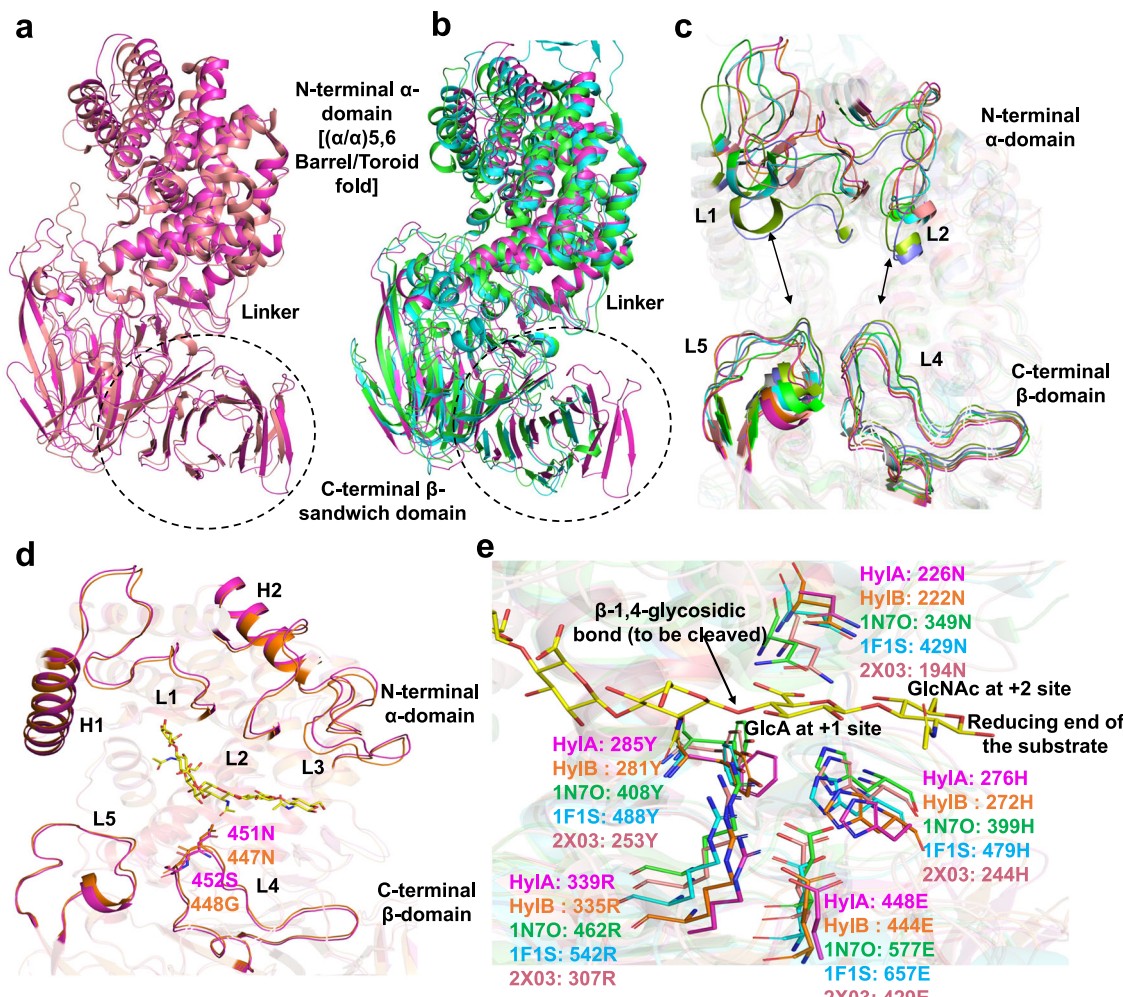

**Fig. 3 | Comparison of HylA and HylB with bacterial and animal Hyl.**
**a** superimposition of HylA (PDB: 8FYG) crystal structure with Hyl from *Streptomyces coelicolor* (ScHyl). **b** comparison of HylA crystal structure with Hyl from *Streptococcus pneumoniae* (SpnHyl) and *Streptococcus agalactiae* (SaHyl). HylA, ScHyl, SpnHyl, and SaHyl are shown by cartoon representation in magenta, salmon red, green, and cyan, respectively. The PDB IDs for ScHyl, SpnHyl, and SaHyl are 2X03, 2BRW, and 1F1S, respectively. The structural domains and linker are labeled. **c** the conformations of the substrate-binding cleft are shown. The relative positions of (i.e., the distances between) the L1 and/or L2 loops from the α-domain and the L4 and/or L5 loops from the β-domain defines the open/closed conformation of the Hyl cleft and is denoted by black arrows. The Hyl enzyme's cleft from different bacteria, including the crystal structures HylA, HylB (PDB: 8FNX), 2X03, 2WCO, 2BRW, 1LOH, 1F1S, and 1LXM are shown by magenta, orange, salmon red, slate blue, green, splitpea green, cyan, and grey70, respectively. **d** HylA and HylB structural elements that define the catalytic cleft are shown in cartoon representation. HylA and HylB are shown in magenta and orange, respectively. The HA–6 ligand is taken from the SpnHyl crystal structure (PDB: 1LOH) and is shown by sticks in yellow. **e** the catalytic tetrad (Tyr-His-Arg-Glu) and residues (Asx) involving in the neutralization of the substrate's acid moiety are shown. The corresponding residues from HylA, HylB, ScHyl, SpnHyl and SaHyl are shown by sticks in magenta, orange, salmon red, green, and cyan, respectively. The HA-6 ligand is taken from the SpnHyl crystal structure (PDB: 1LOH).

measured their hyaluronidase activity to determine which amino acid of each pair is favored at that position. In this assay, the cleavage of HMW-HA by hyaluronidase is recorded by monitoring UV absorbance at 232 nm, which increases with the formation of an unsaturated carbon-carbon bond in the β−D-glucuronic acid moiety at the cleavage site.

This assay shows that HylB degrades HA at approximately twice the rate of HylA, while control mutations of the tryptophan residues of the catalytic triad (HylA/B Y285F/Y281F) severely curtail the HA-degrading activity of both enzymes (Supplementary Fig. 10). For the residue pairs, only HylA/HylB position 346/342 showed a distinct preference, with both HylA and HylB displaying greater enzyme velocity with glutamic acid over glycine. Incongruously, however, the wild-type sequence that contains glutamic acid at this position is not that of HylB but HylA, the less active of the two variants; it is thus unlikely that this residue accounts for part of the difference in cleavage rate between HylA and HylB.

## Mutation of HylA residue Ser 452 to glycine of HylB alters HylA enzymatic phenotype

We further generated point mutations in several residues of HylA to the analogous residues in HylB to recapitulate the HylB phenotype of HA product size. HylA S452G is located in a loop in the C-terminal domain, between strands β10 and β11 (Fig. 3d, Fig. 4a, and Supplementary Fig. 8); mutation of the preceding residue in the analogous loop of *S. pneumoniae* Hyl has been shown to alter enzymatic activity[15,16]. HylA S116E and E346G, as noted above, reside in the catalytic cleft towards the non-reducing end of the HA substrate. HylA S284G lies adjacent to the catalytic Y285, and N442D is placed at an exposed position within the C-terminal domain (Fig. 4a).

HylA S284G, S116E, and E346G had no significant effects on the HylA product size phenotype, though E346G showed decreased overall enzymatic activity and N442D resulted in a nearly complete loss of activity (Fig. 4b–i, and Supplementary Fig. 10 and Supplementary Fig. 11). HylA S452G, however, successfully altered the HylA enzymatic

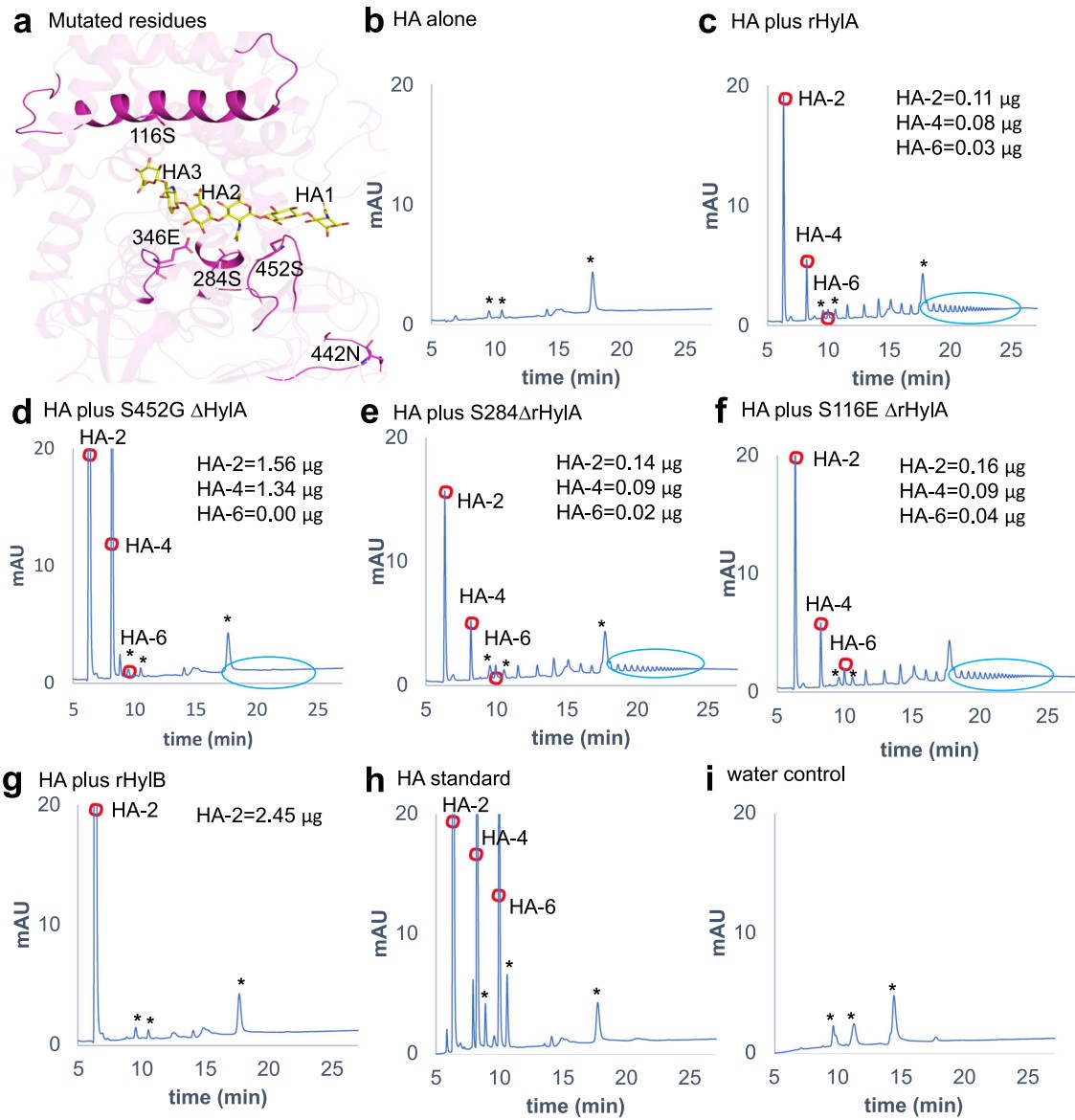

**Fig. 4 | Enzymatic activity of HylA mutants with single amino acid substitutions. a** position of the amino acid residues (shown by sticks in magenta) on HylA (PDB: 8FYG) crystal structure that were mutated to corresponding HylB residues. The HA-6 ligand is taken from the SpnHyl crystal structure (PDB: 1LOH). **b**–**f** HPLC profile of HMW-HA after 24 hr coincubation with WT or mutant HylA (0.35 μg): HA alone (**b**), rHylA (**c**), or rHylA with single amino acid substitutions (**d**–**f**). **g** HPLC

profile of HMW-HA after 24 hr coincubation with WT rHylB (0.35 μg). **h** quantification of HA-digested peaks was performed using known concentrations of purified HA oligosaccharides. **i** water alone run as a blank control. Asterisk (*) in **b**–**i** represents non-specific peaks, present in water control as well. Data are representative of two independent experiments. Source data are provided as a Source Data file.

phenotype; reminiscent of HylB, S452G displayed an increased enzyme velocity and reduced amounts of larger oligomers as product (Fig. 4d). The resulting product size, however, was not predominantly HA-2 as would be expected for a strictly HylB-like phenotype, but was a mixture containing a higher ratio of HA-4 to HA-2 compared to WT HylA. Interestingly, the amino acid residue, S452 in HylA and G448 in HylB, is conserved across *C. acnes* strains (Supplementary Fig. 12) suggesting that a similar hydrolytic process may be conserved.

Earlier domain motions in SpHyl, SaHyl and ScHyl enzymes have been implicated in substrate processing by molecular simulation studies[17,19,20,22,23,26,27,29]. To understand whether substrate processing by HylA and HylB follow a similar mechanism, we performed molecule simulation of HylA and HylB wild-types and HylA-mutants, S452G and E346G as described by Josh et al.[29]. Our molecular simulation study results are consistent with observations of Josh et al.[29] Briefly, PCA analyses of simulated trajectories suggest that HylB-WT is far more

dynamic than HylA-WT and S452G mutation in HylA shows increased (amplitude) domain motions similar to HylB-WT; the cleft opening/closing motion (Eigenvector 1) increased by about 20%, while the other domain motions increased by 10–40% (Supplementary Fig. 13). These observations are consistent with the hypothesis that complex structural dynamics are one of the key mechanisms for substrate processing by HylA and HylB.

## Proinflammatory properties of Hyl degradation products

Having defined the structures and enzymatic functions of *C. acnes* Hyls, we asked if the HA fragments produced by HylA and HylB induce inflammatory pathology noted in the in vivo experiments. For these assays, we digested HA with rHyls or supernatant derived from WT or Δ*hyl* for 24 h and measured cell-specific cytokine secretion responses. We found that HylA-cleaved HA induced higher levels of acne-related cytokines in keratinocytes (Fig. 5a–d) and BMDMs (Supplementary

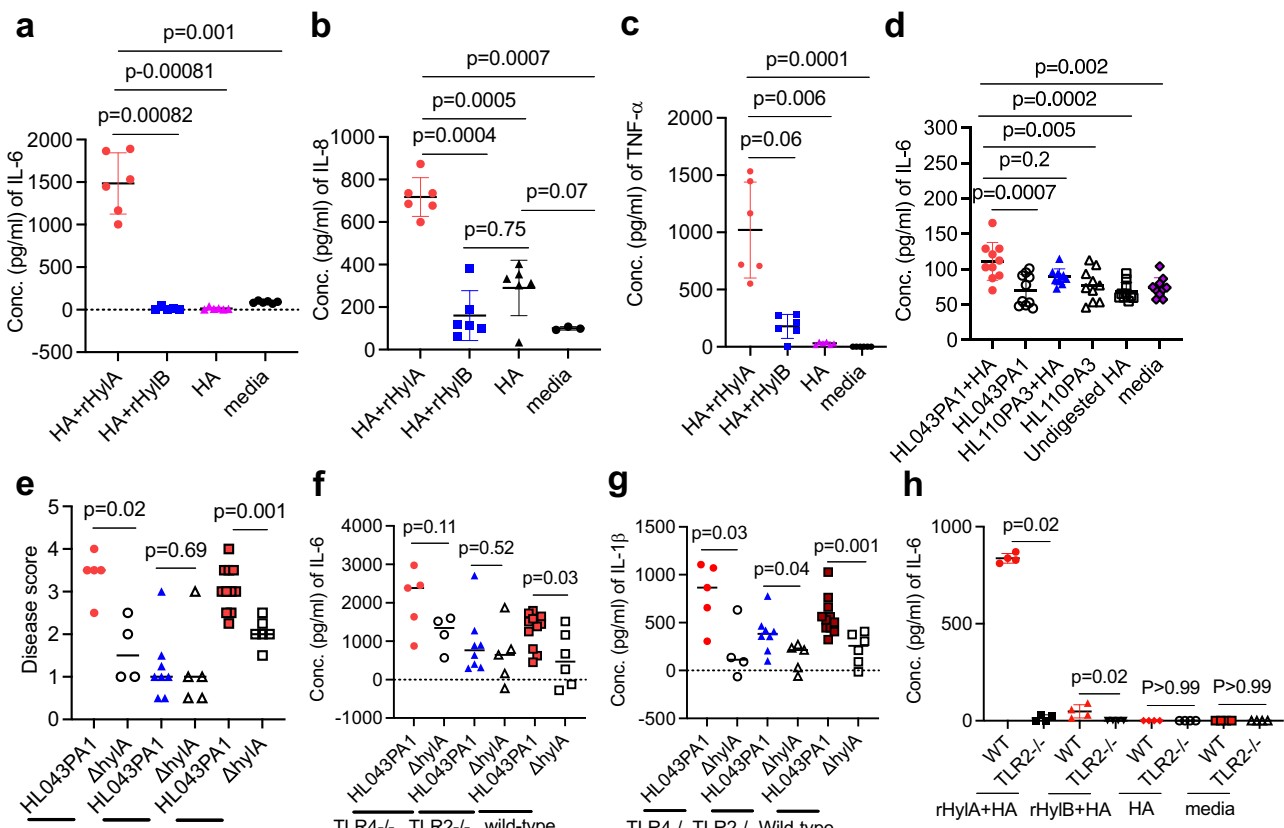

**Fig. 5 | Proinflammatory properties and TLR2 dependence of Hyl degradation products.** HaCaT cells were stimulated with HA, that had been predigested with either rHylA or rHylB, for 24 hr followed by IL-6 (**a**), IL-8 (**b**) and TNF-α (**c**) measurements in the culture supernatant. **d** HaCaT cells were stimulated with the HA, predigested with either supernatant from HL043PA1, HL110PA3 or corresponding isogenic mutant for 24 hr followed by IL-6 measurement in the culture supernatant. WT, *TLR2⁻/⁻* and *TLR4⁻/⁻* mice were infected i.d. with WT or isogenic Δ*hylA* HL043PA1 (2x10⁷CFU) strain as above. Disease score (**e**) and skin cytokines (**f, g**) at 24 hr post-infection. **h** IL-6 in WT or *TLR2⁻/⁻* BMDM culture supernatant after stimulation with rHylA or rHylB digested HA. Data in a (*n* = 5 for HA + rHylA and 6 for other

conditions), **b** (*n* = 3 for media and 6 for other conditions), **c** (*n* = 6), **d** (*n* = 10), **h** (*n* = 4), are presented as mean ± SD and each data point represents one well. The data are representative of two independent experiments. **e–g** Bars denote median, and each data point represents one individual mouse (*n* = 5 *TLR4⁻/⁻*, *n* = 8 for *TLR2⁻/⁻* or *n* = 11 for WT mice infected with HL043PA1, and *n* = 4 for *TLR4⁻/⁻*, *n* = 5 for *TLR2⁻/⁻* or *n* = 6 for WT mice infected with isogenic Δ*hylA*). The *p* values in **a**, **b** were calculated by one-way Welch ANOVA test, *p* values in **c**, **d** were calculated by non-parametric Kruskal-Wallis one-way ANOVA test, and p values in **e–h** were calculated by non-parametric two-tailed Mann-Whitney U test. Source data are provided as a Source Data file.

Fig. 14a) than controls. In comparison, HA degraded by HylB induced almost no change in IL-6 or lower IL-8 and TNF-α levels compared to controls, consistent with the previously defined anti-inflammatory properties of processive HA-2-producing Hyl enzymes. As reported, HA-2 lacks pro-inflammatory property, and degradation of HA to HA-2 abrogates proinflammatory properties of the larger HA fragments[16]. Furthermore, HA-2 competes with the larger-sized HA to further block TLR2 activation[16].

The TLR2 dependence of acne vulgaris is a well-recognized feature of the skin disease[9,30]. Therefore, we interrogated TLR2 dependence of inflammation induced by HylA and HylB. Consistent with the TLR2-dependence of HA, differences in pathology induced by HL043PA1 and Δ*HylA* were abrogated in *TLR2⁻/⁻* but not in *TLR4⁻/⁻* mice (Fig. 5e–h and Supplementary Fig. 14b, c).

Overall, our findings are consistent with Hyl generation of distinct degradation product sizes that leads to different inflammatory outcomes. The potential importance of Hyl in humans is further advanced by linking Hyl mechanisms and acne through their TLR2 dependence.

**Targeting Hyl to treat acne disease**
Above, we have shown that HylA plays a major role in the immunopathology of acne in our murine model. HylA is highly conserved, with consistent enzymatic activity demonstrated across phylotypes of *C.*

*acnes*[10], hence, appears to be a good target for therapeutic intervention. However, the significant homology between HylA and HylB poses a potential challenge of therapeutic selectivity.

To begin, we tested if immunization against HylA conferred protection against acne. We injected mice three times at one-week intervals, then challenged the mice with HL043PA1 Supplementary Fig. 15a). HylA vaccination in alum induced robust antibody response against HylA (Supplementary Fig. 15b) and significantly reduced immunopathology associated with murine acne disease (Fig. 6a, b and Supplementary Fig. 15c, d). However, HylA (Supplementary Fig. 15b) or HylB (Supplementary Fig. 15e) immunization induced cross-reactive antibodies to each other, and modest deterioration of disease score when the mice were challenged with health-associated HL110PA3 strain (Supplementary Fig. 15f–h).

To circumvent the potential for inducing inflammation related to cross-reactive antibodies, we designed a HylA-specific peptide vaccine using the combination of antigenicity program (IEDB analysis resources, http://tools.iedb.org/bcell/) (Supplementary Fig. 16a–d) and the crystal structures of HylA and HylB. The peptide combines several HylA-specific epitopes and has no significant homology to human proteins. We physically linked the predicted peptides to tetanus toxoid (TT), expressed the fusion protein in *E. coli*, and then verified the construct by mass-spectrometry. The vaccine selectively inhibited

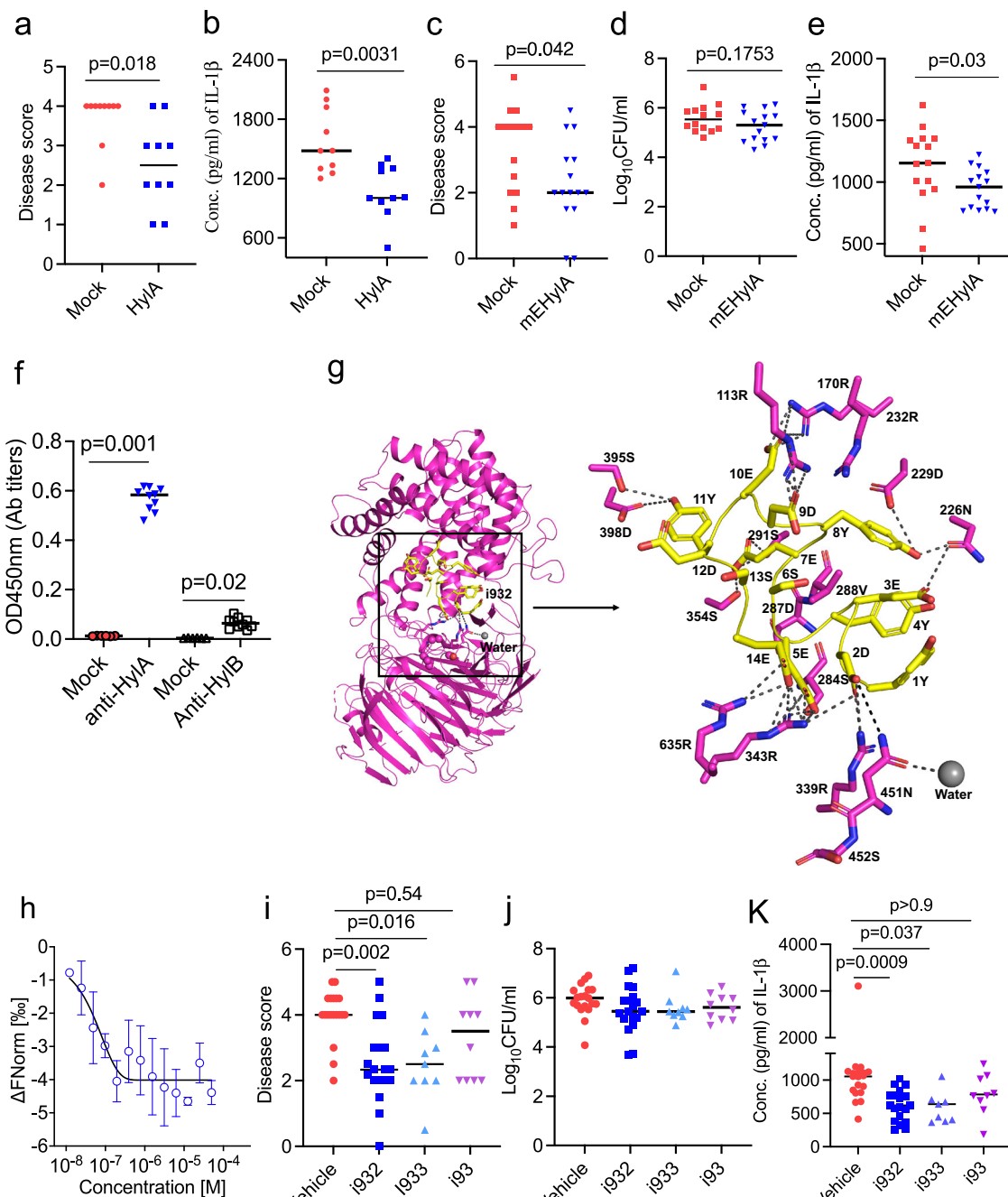

**Fig. 6 | Selective neutralization of HylA improves acne lesions and mitigates inflammation. a, b** CD1 mice ($n = 10$) immunized with either Alum (Mock) or Alum-rHylA (HylA) were challenged i.d. with HL043PA1. Disease score (**a**) and IL-1b in skin homogenate at d2 post-challenge. **c–e** mice ($n = 15$) were immunized intraperitoneally (i.p.) with alum plus C-terminus of tetanus protein (TT) or multiple HylA epitopes linked to TT (mEHylA), then challenged i.d. with HL043PA1 *C. acnes* strain. Disease score (**c**) bacterial burden (**d**), and IL-1b (**e**) at d2 post-challenge. **f** serum (1:100,000 diluted) anti-HylA or anti-HylB IgG antibody titers after the third immunization with mEHylA vaccine. **g**, modeling of the HylA-i932 peptide complex. The i932 peptide docked in the HylA (PDB: 8FYG) active site cleft. The peptide is represented as yellow cartoon with the side chains shown by sticks. **h**, microscale thermophoresis (MST) analysis of HylA binding to peptide i932. MST dose response curve obtained by titrating the i932 peptide (50 μM to 1.5 nM) against 30 nM fluorescent labeled HylA. **i–k**, inhibitors (i932, i933, or i93) at 10 μg and HL043PA1 strain ($2 \times 10^7$ CFU/mouse) were co-injected i.d. into CD1 mice ($n = 19$ for vehicle and i932, $n = 9$ for i933 and $n = 10$ for i93). Disease score (**i**), CFU (**j**), and skin IL-1b (**k**) d1 (24 hr) post-infection. Bars denote median. Data are from two (**a**, **b**, **f**, **i–k**) or three (**c–e**) independent experiments with each data point representing one mouse. Data in **h** is represented as mean ± SD of triplicates of one independent experiment and the experiment was repeated three times. The data in **a–e** were analyzed by non-parametric two-tailed Mann-Whitney U test, and in **f**, **i–k** by non-parametric Kruskal-Wallis one-way ANOVA test. Source data are provided as a Source Data file.

acne caused by HL043PA1 (Fig. 6c–e) with minimal evidence of inflammation from cross-reactivity to HylB from HL110PA3 (Fig. 6f and Supplementary Fig. 17a–f). In an adoptive T cell transfer experiment, CD3+T cells from vaccinated mice were shown to be dispensable for protection against acne disease (Supplementary Fig. 17g–k). Next, we tested the effect of post-mEHylA vaccination serum on rHylA enzymatic activity. Post vaccination serum significantly reduced HA degradation by HylA enzyme (Supplementary Fig. 18a–e). Notably, mEhylA generated predominant IgG1 anti-HylA antibodies (Supplementary Fig. 18f).

In addition to vaccination, we developed selective peptide inhibitors by fragment-based virtual screening using Glide (Schrodinger, Inc. San Diego, CA)[31] based on the structural analysis between HylA and HylB, and tested in vitro (Supplementary Fig. 19a–c) and in vivo (Fig. 6g–k and Supplementary Fig. 20). The inhibitor i932 targeted multiple sites on HylA's substrate binding pocket (Fig. 6g) including the L4 loop containing S452 that was critical for the intermediate HylA / HylB-like enzymatic phenotype. Microscale thermophoresis technique (MST) analysis showed that HylA binds to the peptide i932 with a binding constant (KD) of 21.6 nM (Fig. 6h), while i933 and i93 (Supplementary Fig. 19b, c) had binding constants (KD) of 1.76 μM and 911.5 μM, respectively. As shown in Fig. 6i–k and in Supplementary Fig. 19a and Supplementary Fig. 20a, b, most inhibitors were effective both in vitro and in vivo, with i932 demonstrating the highest efficacy. We verified the lack of effect of the inhibitors on HL110PA3-induced skin disease (Supplementary Fig. 20c–g). Overall, our studies point to the feasibility of structure based, pathogenesis guided, selective approach to the treatment of acne.

## Discussion

Our study supports Hyl as a major virulence factor that explains the divergence of health and acne phenotype of *C. acnes* strains. This is supported by the high degree of association between HylA and HylB with clinical disease or health, their TLR2 dependency in immunopathologic mechanisms, consistent with acne vulgaris, and the contribution of the two Hyl variants to immunopathology and health. Although our data support the prominence of HylA virulence, several other *C. acnes* virulence factors have been reported. These include toxic porphyrin biosynthesis genes that are upregulated with vitamin B12 supplementation[32], and CAMP factor which enlist cytotoxic host sphingomyelinase[33].

Based on our phylogenetic analysis, HylA is the only proinflammatory Hyl elaborated by a human commensal or pathogen. Since *C. acnes* is both a human commensal and a soil bacterium, its clustering among environmental microbes in the phylogeny tree makes sense, perhaps as a transition from soil to commensal. HylA and HylB relatedness to Hyl from soil derived organisms, *Streptomyces* and *Arthrobacter*, can be consistent with this proposed transition from a multi-functioning lyase to a more restrictive and processive enzyme. Because an inflammatory milieu is usually harmful to pathogens, the expectation is that *C. acnes* Hyl would evolve from pro-inflammatory HylA to HylB. While this was anticipated, expression of HylA or HylB do not appear to modify *C. acnes* survival in our acne model to exert a selective pressure on survival. This would be consistent with the finding of abundant *C. acnes* strains that express either of the Hyl. Although HylA and HylB differ by 26% in genomic sequence, we show that one single amino acid substitution can significantly alter the phenotype of the enzyme, suggesting that a major pathogenic potential of *C. acnes* may only be encoded and modified by small changes that occur during the evolution of the enzyme. These single amino acid substitutions are observed to be conserved across the different *C. acnes* strains (Supplementary Fig. 12).

It has been reported that health- and acne-associated *C. acnes* acquired HylB and HylA, respectively, through different insertional events in the indel 14 genomic region, based on finding of different sequences up and downstream of *hylA and hylB*[10,34]. It is unclear how these events occurred, but the structure and sequence relatedness of HylA and HylB compared to other bacterial Hyls suggest that they most likely originated from within Cutibacterium species. Until further genetic data become available, current data are most consistent with that interpretation.

The understanding of the structural differences between HylA and HylB allowed us to devise selective therapeutics that target the proinflammatory enzyme. A report showed that only 4–17% of humans develop neutralizing antibodies to *C. acnes* Hyls and only after early adulthood[35]. Hence, a vaccine approach would be of significant benefit. We acknowledge that an animal model that fully mimics human acne is not available and that findings from our study need to be interpreted with caution in relation to clinical translation. Nevertheless, our approach along with more recently targeted virulence-based therapeutics represent a major initial step in mechanism-based treatment of acne that has largely been missing.

## Methods

### Ethics statement

All animal studies were approved under the guidelines of the University of California San Diego (UCSD) Institutional Animal Care and Use Committee (IRB animal protocol approved number S18200). The mice were housed in an animal facility at UCSD with a standard of care as per federal, state, local, and NIH guidelines.

### *C. acnes* bacterial culture

Two acne-associated strains (HL043PA1 and HL043PA2) and two health-associated strains (HL110PA3 and HL110PA4) were used in this study[6]. Clinical *C. acnes* strains from frozen stock were cultured on blood agar plates anaerobically using BD BBL™ GasPak™ system for 96 h at 37 °C. A single colony of *C. acnes* was anaerobically grown in 10 ml of Brain Heart Infusion (BHI) broth (Catalog no. #53286, Sigma-Aldrich, USA) for 3–4 days (OD = 0.15–0.3), followed by once washing of the bacterial pellet with BHI media at $2300 \times g$ for 5 min. The pellet was resuspended in BHI media to a desired $OD_{600nm}$ (0.5) for in vitro and in vivo studies. Bacterial culture supernatant was collected and used for rooster comb HA (Catalog no. #H5388, Sigma-Aldrich, USA) degradation activity, sodium dodecyl sulfate–polyacrylamide gel electrophoresis (SDS-PAGE) analysis, and human keratinocyte HaCaT cell (ATCC) stimulation.

### Construction of ΔhylA and ΔhylB *C. acnes* strains

The homologous recombination cloning strategy performed was previously described[36] with slight modifications. Briefly, 500 bp up and down stream of the hyaluronidase gene was amplified by PCR (Supplementary Table 3), gel purified, ligated together, and cloned into pGEM-T-easy (catalog #A137A, Promega, USA). The erythromycin resistance cassette from pDCerm was PCR amplified and ligated between the up and down stream regions before being transformed into *E. coli* (DH5α) (catalog # 18265017, ThermoFisher Scientific). Plasmid DNA from ampicillin (100 μg/mL) resistant clones was purified and verified by PCR. Correct plasmids were transformed into dam-negative *E. coli* (Catalog # C2925I, New England Biolabs) and purified. Competent *C. acnes* cells were prepared as described previously[37]. Briefly, *C. acnes* was grown in BHI medium to an $OD_{600nm}$ of 0.5–0.6 anaerobically at 37 °C. Cells were pelleted and washed in EP buffer (272 mM sucrose, 7 mM sodium phosphate, and 1 mM magnesium chloride) twice. Plasmid DNA was mixed with freshly made electro-competent *C. acnes* cells and electroporated. One mL of BHI was immediately added following electroporation. The cells were pelleted and resuspended in 100 μl of BHI medium, plated onto BHI agar plates and incubated at 37 °C anaerobically overnight. The next day the bacteria were removed with a cotton swab, placed in fresh BHI medium, plated onto BHI plates containing erythromycin (10 μg/ml) (Catalog no. #E5389, Sigma-Aldrich) and incubated at 37 °C until colonies appeared (5–7 days). Mutants were verified by PCR and the lack of activity confirmed on agar plates containing hyaluronan.

### Hyaluronidase plate assay

Hyaluronate lyase activity of HylA or HylB in *C. acnes* culture supernatants was measured using HA from rooster comb as a substrate. 20 μL or 40 μL of supernatant from a single colony of *C. acnes*, grown anaerobically for 4 days and harvested at $2600 \times g$ for 10 min, was

spotted on BHI agar plates containing 1% bovine serum albumin (BSA) fraction V (Catalog no. #10735078001, Sigma-Aldrich, USA) and HA (400 µg/mL). The plates were incubated at 37 °C overnight, and HA degradation was detected by flushing the plate with 2 N acetic acid for 3–5 min.

## Cell cultures

Bone marrow-derived macrophages (BMDMs) were isolated from the femurs and tibiae of 12-week-old C57BL/6 mice (Jackson laboratories) and suspended in complete RPMI 1640 media (Gibco, ThermoFisher Scientific, USA) with 10% heat-inactivated fetal bovine serum (FBS), 10 ng/ml of M-CSF (Catalog no. # PeproTech, Inc., USA) and 1% of Penicillin-Streptomycin antibiotics (Catalog no. #P4333, Sigma-Aldrich, St. Louis, MO, USA). Cells were cultured in 92 mm non-adherent dishes (ThermoFisher Scientific, USA) at 37 °C under 5% CO2, followed by the replacement of media with the fresh media containing equivalent concentrations of M-CSF every other two days. Then seven days post-culture, cells were harvested and stimulated with HA (40 µg) that was digested with either bacterial supernatant or rHylA or rHylB enzymes.

HaCaT cells (ATCC) were cultured in complete DMEM media (Catalog no. 10-013-CV, Corning incorporated, USA) plus 10% heat-inactivated FBS in 5% CO2 at 37 °C. Before cell stimulation with digested HA (40 µg), HaCaT cells were seeded in 96-wells Falcon® 96-well tissue culture plate (Catalog no. #353072, corning incorporated, USA) at a concentration of $10^5$ cells/ml and incubated in 5% CO2 at 37 °C for 6 hr, followed by washing with DMEM media and cell stimulation.

## HA digestion for HPLC analysis and cell culture stimulation

HA (2 mg/ml) from rooster comb was digested with either supernatant from *C. acnes* bacterial cultures (10 µl/ml) or purified recombinant protein (rHylA, rHylB or mutant proteins) at a concentration of 0.35 or 1 µg/ml. The digestion was carried out in a reaction buffer containing 100 mM Na acetate, 10 mM $CaCl_2$ and 0.5 mM DTT (pH = 5.5) at 37 °C for different time points (0, 5 min, 15 min, 1 hr and 24 hr), and the reaction was stopped by inactivating the enzyme at 80 °C for 10 min and then stored in −20 °C until further use. Supernatant from bacterial cultures used for HA digestion was 20X concentrated using 50 kDa Amicon® Ultra-15 centrifugal filters (Catalog no. #UFC905024, Millipore Sigma, USA).

For BMDMs and HaCaT cell assays, an equivalent of 40 µg of HA digest was used to stimulate $10^5$ cells for 8 and 16 hr, respectively. Cells were plated in a Falcon® 96-well tissue culture plate and cultured in 200 µl of complete RPMI medium supplemented with 10% FBS and 1X penicillin–streptomycin antibiotics solution. After 24 hr incubation at 37 °C under 5% CO2, the cells were centrifuged at 400 × g and the culture supernatant were collected for analysis of proinflammatory cytokines, including IL-6, TNF-α, and IL-8, by a solid-phase sandwich enzyme-linked immunosorbent assay (ELISA; Biolegend, CA, USA).

HA digested products were analysed by strong anion exchange high-performance liquid chromatography (HPLC), which was performed with the Ultimate 3000 HPLC system (ThermoScientific, USA) equipped with a Ultimate3000 Variable Wavelength Detector on a Pro Pack SAX-10 (4 × 250 mm) column attached to a Pack SAX-10G guard column (4 x 50mm, Thermo-Dionex, USA) at 30 °C. Two different solvents were used; Solvent-A (HPLC-water pH 3.5) and Solvent-B (2 M NaCl, pH 3.5) at flow rate of 1 mL/min. The gradient conditions (linear) are mentioned in Supplementary Table 4. The chromatogram was acquired with UV absorbance set at 232 nm. A known amount of sample was dissolved in UP water and injected on HPLC. Standard mixture of 1 µg each of HA-DP2, HA-DP4 and HA-DP6 was injected and the HA oligosaccharides in the samples were quantified by comparing the area under the peaks with the standard mixture.

## Expression and purification of recombinant enzymes

*C. acnes* HylB (residues 37-801) and HylA (41-805) were cloned into pET His6 TEV LIC (Catalog no. # 29653, Addgene) and pET His6 MBP TEV LIC (Catalog no. # 29656, Addgene) cloning vectors, respectively (Supplementary Table 3), and propagated in *Escherichia coli* Top10 cells (Catalog no. # C404010, ThermoFisher Scientific, USA). The recombinant plasmids were transformed into *E. coli* BL21(DE3) pLysS cells (Catalog no. #C606010, ThermoFisher Scientific, USA), and the protein expression was induced by addition of 0.1 mM IPTG (Catalog no. #16758, Sigma-Aldrich, USA) to bacterial cultures (OD = 0.6 nm), followed by incubation of cultures at 18 °C for 16 hr. Then bacteria were pelleted at 17700 × g for 10 min and the pellet was resuspended in lysis buffer (50 mM $Na_2HPO_4$, 300 mM NaCl, 2 mM $MgCl_2$, 10 mM imidazole, 1% Triton X-100, 1 mg/ml egg white lysozyme, 1 mM PMSF, and 10 µg/ml DNases; pH 8). The bacterial lysate was stored in −80 °C for 24 h, followed by freeze-thawing at 4 °C and centrifugation at 17,700 × g for 20 min. The supernatant was harvested and incubated with His60 Ni Superflow™ resin (Catalog no. #635660, Takara Bio USA, Inc.) for 4 hr, followed by passing the mixture through chromatographic gravity columns. The resin was washed thrice (total 90 ml) with wash buffer (50 mM $Na_2HPO_4$, 300 mM NaCl and 25 mM imidazole; pH 7.4), and the protein was eluted by 15 ml of elution buffer (10 mM $Na_2HPO_4$, 300 mM NaCl and 300 mM imidazole and 0.1% Tween 80; pH 7.4). The eluted protein was washed thrice with PBS-T buffer containing 0.1% tween-80 using 50 kDA Amicon™ centrifugal filters. The purity of purified proteins was confirmed by SDS-PAGE analysis. The purified proteins were cleaned off from LPS contamination using Pierce™ high-capacity endotoxin removal spin columns (Catalog no. # 88274, ThermoFisher Scientific, USA) following the instructions of manufacturer.

Mutant *hylA* and *hylB* constructs were cloned and expressed as above. (Supplementary Table 5). The concentrations of proteins were estimated by NanoDrop 2000 Spectrophotometer (ThermoScientific, USA), and stored in −80 °C until further use.

For protein crystallization studies, BL21 bacterial cells harboring HylA and HylB proteins were harvested by centrifugation at 4000 × g for 15 min, followed by resuspension in Ni buffer (30 mM HEPES, 500 mM NaCl, 10% glycerol, 20 mM imidazole, 5 mM β-mercaptoethanol; pH 7.5) supplemented by Roche Complete EDTA-free protease inhibitor cocktail. Samples were lysed by sonication, centrifuged at 23,700 × g for 40 min to remove cellular debris, and applied to a HisTrap FF crude column (GE Healthcare). Protein was eluted using Ni buffer containing 500 mM imidazole. After incubation with TEV protease and dialysis into Ni buffer overnight, samples were again applied to a HisTrap FF crude column to remove uncleaved product. Samples were then purified over a Superdex 200 Increase 10/300 GL column (GE Healthcare) into buffer containing 100 mM Na acetate pH 5, 10 mM $CaCl_2$, and 0.5 mM TCEP, concentrated, and frozen with liquid nitrogen.

## Molecular dynamics simulations

The molecular dynamics simulations were carried out using the GROMACS software package version 2022.4[38] and as described by Joshi HV et al.[29]. Briefly, the HylA-Y285F and HylB-WT apo crystal structures were modeled with missing residues and the residue Phe285 was mutated back to Tyr285. Then two mutant HylA (S452G and E346G) models were generated. Four models (HylB-wt, HylA-wt and HylA-mutants) were used as starting models for simulation studies. A 100 ns MD run was carried out for all four simulations done in this study. Domain motions (Eigenvectors) for each model were determined using PCA analysis from Gromacs package (https://manual.gromacs.org/2022.4/manual-2022.4). The cleft opening/closing motion (Evec1) was determined as the Cα-Cα separations of Ser97 and Thr636 (HylA numbering); the domain twisting motion (Evec2) as the Cα-Cα separations of Glu208 and Pro216; the substrate-entry

opening/closing motion (Evec3) as Cα-Cα separations of Thr80 and Thr636, and the product-exit opening/closing motion (Evec4) as Cα-Cα separations of Thr80 and Thr636.

## HylA peptide inhibitor design

Peptide inhibitors of HylA were developed by structure-based virtual screening using Schrodinger software package version 2021-1 (Morin et al.[39] Schrodinger, inc. San Diego, CA) Briefly, the X-ray crystal structure of HylA was prepared with the Protein Preparation Wizard in MAESTRO version 12.8. During the protein preparation, the bond orders were assigned, and hydrogen atoms and formal charges were added to hetero groups. The water molecules in the ligand-binding area were preserved for docking, and all other water molecules 5 Å beyond hetero groups were deleted. The hydrogen bonding network of binding site residues was optimized by selecting the histidine tautomers and by predicting the ionization states. The prepared HylA structure was used for the molecular docking simulations. For virtual screening, a library was built with 200 fragments available at Chembridge Corp (San Diego, CA) and 98 amino acid fragments (L, D and non-natural) from Enamine.net. using ligprep module. The prepared fragment library of molecules was docked flexibly utilizing GLIDE in XP mode. For ease of synthesis, amino acid fragments that dock in the substrate binding pocket with high glide scores were selected and linked with other amino acid residues as peptides. About 20–30 peptides were designed of different lengths. The designed peptides were docked against HylA using the protein-protein docking method. Finally, 8–10 peptides were selected for binding assays.

## Design of the HylA multi-epitope vaccine

Linear B cell epitopes within HylA protein were predicted using Bepipred Linear Epitope Prediction 2.0 (IEDB analysis resources, http://tools.iedb.org/bcell/). Immunogenic peptides with scores above 0.5 were selected and aligned with HylB. Four peptides with no homology to HylB were selected and physically linked to the C-terminus of a tetanus toxoid protein. Linker amino acid glycine (G) was placed between each peptide and the C-terminus of tetanus toxin[40] (see Supplementary Fig. 16). The fusion gene (mEHylA) was then optimized for *E. coli* protein expression, cloned into pET28a (+) (GenScript), and transformed into *E. coli* BL21(DE3) pLysS cells. The HylA multi-epitope construct was then expressed, purified as described above, and assessed for accuracy of sequence by mass spectrometry.

## Crystallization, data collection, and structure determination

Crystals of HylB were grown using the hanging drop vapor diffusion method by adding equal volumes of protein and reservoir well solution (0.2 M Na dihydrogen phosphate pH 6.5, 9% PEG 8000, 5 mM TCEP) and suspending over well solution at 18 °C. Crystal size and quality were improved using streak seeding. Crystals were cryoprotected in well solution that contained 12% PEG 8000 and 25% glycerol, and flash-frozen in liquid nitrogen. Crystals of HylB Y281F were grown as above using well solution 0.1 M bis-tris pH 6.5, 0.4 M MgCl$_2$, 16% PEG 3350, and 5 mM TCEP and cryoprotected in well solution that contained 20% PEG 3350 and 25% glycerol. Crystals of HylA Y285F were grown and cryoprotected as above for WT HylB except that 0.1 M Na dihydrogen phosphate pH 6.5 was used. Diffraction data were collected at 100 K temperature and 1.5418 Å wavelength on a Rigaku MicroMax-007HF rotating anode X-ray generator with R-Axis IV++ detector.

Diffraction data were processed using XDS[41], and scaled using Scala[42] (HylB WT and Y281F) and XSCALE[41] (HylA Y285F). Molecular replacement for WT HylB was performed using PHASER[43], with the N-terminal domain of *S. agalactiae* hyaluronate lyase (PDB: 1F1S)[21] and the C-terminal domain of *A. aurescens* chondroitin AC lyase (PDB: 1RWA)[24] as search models. The structure of WT HylB was used as a

search model to solve HylB Y281F and HylA Y285F. Model building and refinement were performed using COOT[44] and PHENIX[45]. After refinement, the Ramachandran statistics for the HylB WT are 97.6% favored, 2.4% allowed, and 0% outliers; while it is 96.82%, 3.11%, and 0.07%, respectively, for HylB Y281F; and 96.5%, 3.37%, and 0.13%, respectively, for HylA Y285F. The structural figures were prepared using the PyMOL visualization tool (The PyMOL Molecular Graphics System, Version 2.4 Schrödinger, LLC.). All the above crystallographic and structure visualization & analysis tools/applications were used on the SBGrid Consortium platform [www.sbgrid.org][39]. Root mean square deviations between crystal structures of GAG lyases were performed using the Dali Server[46]. The crystal structures and associated data are available from the Worldwide protein data bank (wwPDB). The PDB codes for HylA (8FYG) and HylB (8FNX, 8G0O).

## Phylogenetic tree building

HylA and HylB FASTA amino acid sequences were obtained from NCBI or the Worldwide protein data bank (wwPDB) and a phylogenetic tree building was performed using Geneious prime software.

## Hyaluronidase enzyme assay

HylB at concentration 15 ng/mL and high molecular weight hyaluronic acid (HMW-HA) at concentration 0.2 mg/mL were added to a 96-well UV-Star clear microplate (Greiner Bio-One, #655801). Reactions were monitored over 10 min at wavelength 232 nm using an Infinite M200 Pro UV spectrophotometer (Tecan). Reaction volume was 100 μL. Assay buffer contained 100 mM Na acetate pH 5.5, 10 mM CaCl$_2$, and 0.5 mM TCEP. Reaction velocities (absorbance units/sec) were obtained using the slope calculated by Magellan software v. 7.0 over reaction time 1–9.5 minutes. All reactions were performed in triplicate. HMW-HA was hyaluronic acid sodium salt from rooster comb, Sigma #H5388, MW 1–4 million Da. This method was adapted from a previous work[47].

## Microscale Thermophoresis (MST) measurement of the HylA binding to peptide inhibitors

The Monolith NT.115 instrument (Nanotemper Technologies, München, Germany) was used to measure the HylA binding to the peptide inhibitors, including i93, i932, and i933. For the MST experiment, the HylA is fluorescently labeled by cysteine labeling using the Monolith protein labeling kit RED-MALEIMIDE 2nd generation (Catalog Number: L014). The assay was performed in 100 mM Sodium acetate buffer pH 5.3 supplied with 0.05% Tween-20. A serial dilution of the individual peptides were titrated against the fluorescent labeled HylA. For the titration, 10 μL of each concentration of the peptide was mixed with 10 μL of the fluorescent labeled HylA. The MST measurement settings include high MST power, 100% excitation power, Nano-RED excitation type and 25 °C thermostat setpoint. Affinity for the peptides' binding to HylA was determined by fitting the MST response data to the KD model using the MO.Affinity Analysis Software version v2.3* (Nanotemper Technologies, München, Germany).

## Mouse acne model

Six weeks-old C57BL/6, *TLR2*$^{-/-}$ (Strain #:004650), and *TLR4*$^{-/-}$ (Strain #:004650) mice were purchased from Jackson Laboratories. *TLR2*$^{-/-}$ and *TLR4*$^{-/-}$ mice were bred in specific-pathogen free facilities. Outbred 6 weeks-old female CD1 mice were purchased from the Charles River Laboratory. All mice (Mus musculus) were kept in filter-top cages with access to food pellet and water under controlled ambient temperatures (20–22 °C), relative humidity (30–70%) and 12 h light/12 h dark cycle. The animal experiments were performed at approximately 8 weeks of age unless otherwise specified. For preparation of BMDMs, 12 weeks-old mice were used.

To model human acne disease, 8 weeks-old mice were i.d. infected with *C. acnes* strains (2 x 10$^7$ CFU in 50 μl volume of BHI media),

followed by the topical application of synthetic sebum daily as described previously[7]. i.d. infections were performed under vaporized Isoflurane (Fluriso, Vet One) anesthesia. Synthetic sebum was made by mixing fatty acid (17% oleic acid; Catalog no. #O1008, Millipore Sigma), triglyceride (45% triolein; Catalog no. # ICN10312201, FisherScientific), wax monoester (25% jojoba oil, Trader Joe), and squalene (13%; Catalog no. # AC215351000, FisherScientific). One (24 hr) or two days (48 hr) after infection, disease score was assessed and the mice were euthanized by $CO_2$. Skin lesions were aseptically excised and harvested in phosphate buffer saline (PBS, pH 7.4). The skin lesions were then homogenized and 25 μl was serially diluted (10-fold) in PBS to determine CFU on BHI agar plates. The BHI agar plates were incubated anaerobically at $37^0C$ for 3–4 days. In addition, homogenized skin lesions were centrifuged at maximum speed ($15870 \times g$) for 25 min and the supernatant was collected and stored in −80 °C for additional analyses.

Disease scoring: Gross skin pathology was scored based on tabulation of the following: Erythematous change (no = 0, mild = 1, moderate 2, and marked = 3); papule (flat = 0, slightly raised = 0.5, small = 1, large = 2, and extra-large=3) based on a protocol modified from[7].

### Mouse immunization
Eight-weeks-old CD1 mice were vaccinated i.p. with 200 μl of alum, alum-rHylA, alum-rHylB or alum-tetanus toxoid-multi-epitope HylA fusion protein (mEHylA) at day 1, 7 and 14. Alum-rHylA and Alum-rHylB were prepared by mixing rHylA or rHylB enzyme with 500 μg of Alhydrogel® Alum adjuvant (Catalog no. #vac-alu-50, InvivoGen), followed by gentle rocking on ice for 1 hr. The vaccines were dosed at 70 μg for the first injection and 50 μg for the subsequent two injections. Serum samples were collected seven days after the last vaccination to assess antibody titers against rHylA or rHylB. To assess the protective effect of vaccination against acne, two weeks post-vaccination mice were challenged i.d. with the clinical HL043PA1 or Hl110PA3 *C. acnes* strains ($2 \times 10^7$CFU). Bacterial count (CFU/ml), size of the skin lesions and proinflammatory cytokines were determined on day 1 and 2 post-challenge as previously described[7].

### Adoptive transfer of T cells
Spleens collected on d10 after the last vaccination were homogenized in sterile PBS (pH 7.4), followed by RBC lysis (Catalog number: 00-4300-54, eBioscience™) and isolation of $CD3^+T$ cells (Catalog No.# 480031, Biolegend, USA) by negative selection using MojoSort™ Mouse CD3 T Cell Isolation Kit as per the manufacturer's instructions. $1 \times 10^7$ $CD3^+T$ cells were retro-orbitally injected into naïve recipient mice. The retroorbital injection was performed under Isoflurane (Fluriso, Vet One) anesthesia. 20 hr post-cell transfer, mice were challenged i.d. with HL043PA1 ($2 \times 10^7$CFU), followed by CFU determination, disease score and skin cytokines on d2 as described above.

### Serum neutralization of HylA enzymatic activity
HylA enzyme (0.3 μg) was incubated at 37 °C for 20 min with 10 μl of pooled serum ($n = 5$), isolated from either mock or mEHylA vaccinated mice. HA (2 mg/ml) was added to the mixture and incubated for 20 hr at 37 °C under continuous rocking. Subsequent HPLC analysis was carried out as mentioned above.

### Studies using specific inhibitors of HylA
Peptide inhibitors specifically targeting HylA enzymatic activity were designed based on the crystal structure of HylA and synthesized (Cedars-Sinai core facility). Inhibitors were tested both in vitro and in vivo.

For in vitro assays, inhibitors at 5 and 10 μM concentrations were tested to block the HA degrading activity of rHylA. Reaction containing HA (2 mg/ml), rHylA (1 μg/ml) and inhibitor (5 or 10 μM/ml) were incubated at 37 °C for 24 hr followed by heat inactivation of an enzyme at 80 °C for 10 min. 20 μl of the resultant mixture was used to stimulate HaCaT cells to measure proinflammatory cytokine IL-6 by ELISA. rHylA plus HA was used as positive control in this assay.

For in vivo experiments, inhibitors (10 μg) were injected along with *C. acnes* strains ($2 \times 10^7$CFU) i.d. into CD1 mice, followed by the topical application of sebum. After 24 hr, bacterial count (CFU/ml), size of the skin lesions and proinflammatory cytokines (IL-1β, IL-6 and TNF-α) in skin lesions were measured.

### Determination of cytokines in skin lesions
IL-1β, IL-6, and TNF-α cytokine levels in skin homogenates, previously stored at −80 °C, were measured by a solid-phase sandwich ELISA using commercially available mouse cytokine ELISA kits (Biolegend, San Diego, CA, USA). The assay was performed in biological replicates as per manufacturer's instructions. The skin homogenates (50 μl) for IL-1β and IL-6 were diluted 1:1 with the blocking buffer (1% BSA plus 1XPBS-Tween20) and undiluted skin homogenate (100 μl) for TNF- α were used in the assay along with the known concentration of cytokine standards (provided with the kits) in each ELISA plate. The plates were developed and read at optical density (OD) of 450 nm with a wavelength correction set to 570 nm in a multimode microplate reader (PerkinElmer, Waltham, MA, USA). The standard curve generated from the OD of cytokine standards was used to determine cytokine levels in the samples. For determination of cytokine levels in culture supernatants of HaCaT cells, human IL-6 and IL-8 cytokine ELISA kits were purchased from Biolegend. Culture supernatant was diluted 1:1 and the assay were performed as mentioned above.

### Determination of antibody titers
Serum antibody titers against rHylA and rHylB were measured by an indirect ELISA method. Briefly, 96-wells high binding microplate (Catalog no. #655081, Greiner Bio-one) were coated overnight at 4 °C with a 500 ng of either rHylA or rHylB in carbonate-bicarbonate buffer (0.2 M, pH 9.6). The plates were washed thrice with PBS-T and blocked with 1%BSA (dissolved in PBS-T buffer) for 1 hr at room temperature under continuous rocking. Following washing, the sera samples (1:100, 1:1000, 1:10,000, and 1:100,000) diluted in PBS-T were applied to the wells in biological replicates and the plate was incubated for 2 hr at room temperature under continuous rocking. The antibody titers (IgM, IgG, IgG1, IgG2b and IgG3) were detected using goat anti-mouse HRP conjugated antibodies (SouthernBiotech) at 1:5000 dilution in the blocking buffer for 1 hr at room temperature under continuous rocking. After step wash (thrice), the plates were developed at room temperature for 3 min using TMB substrate (Catalog no. #555214, BD Bioscience, CA, USA) and the reaction was stopped by addition of 1 N $H_2SO_4$. The plates were read at optical density (OD) of 450 nm with a wavelength correction set to 570 nm in a multimode microplate reader (PerkinElmer, Waltham, MA, USA).

### Statistical analysis and data reproducibility
GraphPad prism version 8 was used to analyze all data (GraphPad Software, San Diego, CA, www.graphpad.com). Specific statistical analyses were noted in the figure legends. In vitro experiments were performed independently 2–3 times with at least three technical replicates and data were presented as mean ± standard deviation. In vitro data were analyzed by non-parametric unpaired two-tailed Mann–Whitney U test for comparison of two groups, and One-way ANOVA for comparison of more multiple groups. All the in vivo mouse data were presented as median of two or more independent experiments. Two-group analysis used a non-parametric unpaired two-tailed Mann–Whitney U T test. Comparisons of multiple groups were performed using one-way ANOVA with Tuckey's post-hoc test. In the case of missing normality, non-parametric Kruskal-Wallis one-way ANOVA was used to analyze the data. For unequal sample size in vitro data,

one-way Welch ANOVA was applied. Higher disease score and inflammation induced by HL043PA1 in comparison to Δ*hylA*, HL110PA3 and Δ*hylB* was confirmed by two independent researchers (IAH and MLD). Vaccination with rHylA enzyme was performed by two independent researchers (IAH and SK).

### Reporting summary

Further information on research design is available in the Nature Portfolio Reporting Summary linked to this article.

## Data availability

Any data generated or analyzed during this study, associated protocols, materials within the manuscript and public databases (PDB database) are included in the article and also available from the corresponding authors upon request. The authors have filed a patent application for HylA and HylB bacterial mutants, recombinant HylA and HylB proteins, inhibitors targeting HylA (i93, i932 and i933), and a vaccine construct (mEHylA), and these materials will be available upon approval of the patent application or with specific request made to the authors prior to patent application approval. Source data are provided with this paper. Structural data from crystallographic studies are available from Worldwide protein data bank (wwPDB): the PDB codes for the crystal structures HylA (8FYG) and HylB (8FNX, 8G0O). Source data are provided with this paper.

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

## Acknowledgements

This study received funding from the National Institutes of Health (R01AI141401 To GYL and RM, and R21AI127406 to GYL). NIH has no role in designing the experiments and the content of this manuscript is the sole responsibility of authors.

## Author contributions

I.A.H. performed most of mouse in vivo experiments with support from C.G. I.A.H. and M.L.D. performed in vitro experiments with support from C.G. and J.R.C. S.K. generated HylA and HylB knock strains. X.D. and T.D. helped with the protein purification. I.A.H., M.L.D., M.P. and B.C. performed HPLC experiments. B.C. and C.M.T. helped with mass-spectrometric analysis of HA disaccharides. M.L.D. performed phylogenetic analysis with support from H.L. I.A.H. and M.L.D. analyzed the data. M.K., R.M.N. (Randall McNally) performed crystallographic, biochemical, and mutational studies; M.K. performed molecular simulation studies; M.K. and A.C. performed peptide inhibitor binding studies guided by R.M. G.Y.L., R.M., I.A.H., M.K. and M.D.L. wrote the manuscript with input from all co-authors. G.Y.L. and R.M. conceived the study. All the experiments were guided by R.M. and G.Y.L.

## Competing interests

R.M., G.Y.L., I.A.H. and S.K. have filed for a patent application for the use of selective HylA inhibitors and vaccines as therapeutics. The remaining authors declare no competing interests.
