## [Peer Review File · Nature Communications]

Functional divergence of a bacterial enzyme promotes healthy or acneic skinREVIEWER COMMENTS

Reviewer #1 (Remarks to the Author):

This manuscript is very interesting for showing the relationship between *Cutibacterium acnes* phenotype and pathogenicity of acne vulgaris. Especially, difference of degradation activity of hyaluronidase is well discussed. Many data was showed in this manuscript for the publication in this journal. However, I need you to submit additional informations as follow.

1. Page 4, Line 81. and Supplementary Table 1.

Strain HL043PA1 was grouped into IA-1, RT5 and ribotypes of many strains were showed (PLoS ONE 9(8): e104199). Text should be corrected and major ribotype in suppl table 1 should be reviewed.

2. The degradation activity of HylA and HylB differed by amino acid substitutions and amino acid sequences of HylA and HylB differed by strains and its activity altered. Is there possible for that mutated HylA or HylB show degradation activity of like HylA or HylB, respectively.

3. Extended data Fig. 10.

Do same genotype strains have same HylA and HylB with mutations? If an enzyme activity is dependent for pathogenicity of acne, strain's genotypes and enzyme activity may roughly match.

4. Fig. 1a.

Was the strains and genetic data used from previous study? Please add appropriately reference and show the detailed data (no. of strains, strain name...).

Reviewer #2 (Remarks to the Author):

The authors convincingly demonstrate the ability of HylA enzyme to cause acne when expressed by *C. acnes* bacteria, in contrast to HylB enzyme. They further clearly show that this is because HylB efficiently digests hyaluronan into disaccharides whereas HylA produces a mix of larger oligomers. As such, the degradation products of HylA activate TLR-dependent inflammatory processes, in contrast to the non-inflammatory disaccharides produced by HylB. The manuscript is additionally strengthened by the x-ray crystal structures of the two enzymes, and the discovery of peptide inhibitors of HylA. All of these factors contribute to my general enthusiasm about the manuscript. I do feel the manuscript could be strengthened by addressing the following comments:

1. What proportion of acne cases is estimated to be caused by HylA producing *C. acnes*?

2. The authors' efforts to derive enzymatic mechanism from their crystal structures are not convincing, and statements that should more properly be put forth as hypotheses are instead put forth as statements of fact. For example:

a. "Hence, replacing Ser to Gly in HylA at [location 452] imparts higher flexibility in loop L4 and increases the domain motion." No data on HylA/B are provided to support this. Rather it is assumed this will be true based on another group's work on a homologous protein.

b. Based on the above logic, mutations that introduce glycine impart flexibility, which in turn enables "accelerated HA degradation." It is therefore surprising to me that, "E346G that decreased the enzymatic activity of HylA." Does this finding contradict the above logic?

c. "the major differences between HylA and HylB, which include 346E/342G, 394A/390S, 395S/391T, 442N/438D, and 452S/448G (Extended data Fig. 11). These differences can critically alter the HA degradation mechanism, as the residues involving in the domain movements and structural flexibility can regulate the substrate entry and translocation/sliding between the subsequent catalytic cycles of the processive degradation of the polymeric/oligomeric HA substrate." This conclusion is based upon an extrapolation from another group's work on a homologous protein. Therefore, the tone of certainty of this conclusion does not seem justified.

d. "The observation that HylA cleft is more open than the HylB cleft (Supplementary Tables 6 and 7) is in correlation with the above inference of mechanistic differences in their HA-degradation." It

is not clear to me what the author's justification for this statement is. That is, I did not see them present a convincing argument that the difference in openness of the cleft is a meaningful correlation with the assumed mechanistic differences.

e. "For the processive nature of degradation, HA binding must be well balanced to be strong enough to allow for HA to remain in the cleft after catalysis and weak enough to allow for sliding of the HA in the cleft for the next round of the reaction. However, the dominant electropositive charges of the HylA cleft result in very strong binding of HA that makes the sliding/translocation of the HA very difficult between the subsequent catalytic events of processive exolytic degradation. While the balanced positive and negative electrostatic charges in the HylB cleft allow the proper binding and sliding of the polymeric/oligomeric HA substrate between the subsequent catalytic events." It seems to me that the authors do not provide any of their own data to back these statements. If this is indeed the case, I would urge them to rephrase this statement to emphasize that this is a hypothesis based on structural differences between their own apo crystal structures, and, perhaps, based on others' work on homologs. If the homolog work is indeed relevant, it would be useful to cite relevant findings from that work directly adjacent to this statement.

f. "These differences may contribute largely to the different amplitudes of the domain motions, with higher magnitudes in HylB making it to act predominantly by exolytic processive manner to yield only disaccharide products, and lower amplitudes in HylA rendering it to exploit the exolytic processive mechanism to lesser extent and thereby yielding multiple oligosaccharide products including HA-4 and HA-6 along with HA-2." Are there data supporting differences in domain motion amplitudes between HylA and HylB, or is this a hypothesis that would need to be tested?

3. "Peptide inhibitors specifically targeting HylA enzymatic activity were designed based on the crystal structure of HylA" Please provide some detail around this statement. Discovery of selective peptide inhibitors of an enzyme that degrades a carbohydrate polymer is not a trivial undertaking, and I would therefore be interested in learning how the authors achieved this success.

4. "The i932 peptide docked in the HylA active site cleft." Please add relevant information on docking to the Methods.

5. "These structural regions in the homologous enzymes, including Schyl (PDB: 2X03), SpnHyl (PDB: 2BRW) and SaHyl (PDB: 1F1S), were shown to be involved in substrate attraction, binding, positioning and translocation, and product release." If allowable under journal guidelines, please cite relevant references for this statement that is part of the caption for Extended data Fig. 9.

6. Sentence structure and flow are of variable quality. In places, especially toward the beginning of the manuscript, they are excellent. However, in certain subsequent sections this quality is noticeably less.

Reviewer #3 (Remarks to the Author):

This is a very well-written paper from a leading group in *C. acnes* research. The link of hyaluronidase variants exclusively expressed by *C. acnes* strains, demonstrate remarkable clinical correlation with acne. This is novel and exciting, and the data are convincing. This is an important contribution to our field and provides further support for targeting HylA as an approach for acne therapy. However, although the topic is of general interest for the readers of the *Nature Communications*, the *in vivo* animal results are preliminary and over-interpreted. The premise for the immunization experiments is unclear as currently, there is no single animal model that encompasses all the pathogenic factors and their interactions in acne formation, which is a genuine human disease. The overarching question is whether the immunization experiments in mice are physiologically relevant in human acne? In addition: i) The immune correlates of protection in the HylA immunization experiments are not defined—which begs the question as to the physiological relevance of these experiments and if acne is a systemic disease; ii) The antibody titers and the classes of antibodies induced by the HylA injections are also not defined iii) The T cell responses induced after injection and their role in protection is not mentioned or discussed iv) There is no rationale for the design and use of the HylA multi-epitope vaccine in the experiments; why only B cell epitopes without consideration for T cell epitopes as well and iv) It is unclear if the selected B epitopes would induce cross reactive antibodies with native human proteins. Overall, the interpretation of data especially the HylA immunization experiments should be revised accordingly as they distract from an otherwise well-designed study.

Additional comments to the authors

1. It is not clear when the cytokine levels were measured post i.d immunizations?
2. In Fig. 5, important controls are missing for example i) non-stimulated controls are lacking ii) Cytokine levels in animals injected with sebum vs. sebum + *C. acnes* HylA and HylB and/or HaCAT cells stimulated with Sebum alone
3. Other cytokines such as IL-8 and TNF- α levels should be measured as well
4. The introduction is not comprehensive and key papers in the field are not cited. the following should be cited (Page 3-line 50-51 work by Yu et al PMID: 27377696, Agak et al PMID: 28864077); Pg 3 line 64 (Kim J et al 2002, PMID 12133981); Pg 4 line 76 (Yu et al. PMID: 27377696)
5. There is need to show proof that HylB is anti-inflammatory: the authors need to measure the levels of anti-inflammatory cytokines e.g., IL-10 (Pg 5 line 99-100)
6. Page 5 line 106 The 2013 publication is not recent. Please edit accordingly. Page 11-line 225 fix typos
7. Page 11-line 240 key papers in the field should be cited (Kim J et al 2002, PMID 12133981)
8. The authors should make the section headings consistent
9. Some supplemental figures need edits
10. Ext. data fig. 6 Hyl expression by 43PA2 and 43PA1 is negligible via WB. Is there a better way to quantitate this ?
11. Ext. data fig. 7 needs caption
12. Ext. data fig. 8 yellow rectangle not visible. Please select a different color
13. What is the green circle explanation in ext. Data fig. 12?
14. Table 1: Please document that some IB phylotypes express HylB
15. It would upvalue the manuscript to include T cell data regarding the vaccine experiments.

Reviewer #1 (Remarks to the Author):

This manuscript is very interesting for showing the relationship between *Cutibacterium acnes* phenotype and pathogenicity of *acne vulgaris*. Especially, difference of degradation activity of hyaluronidase is well discussed. Many data was showed in this manuscript for the publication in this journal. However, I need you to submit additional informations as follow.

1. Page 4, Line 81. and Supplementary Table 1.

Strain HL043PA1 was grouped into IA-1, RT5 and ribotypes of many strains were showed (PLoS ONE 9(8): e104199). Text should be corrected and major ribotype in suppl table 1 should be reviewed.

Response: Here we used the phylogenetic clade notation based on the original publications of the whole genome comparisons (Journal of Investigative Dermatology 2013; doi:10.1038/jid.2013.21 and mBio 4(3):e00003-13. doi:10.1128/mBio.00003-13), where HL043PA1 belongs to Clade IA-2, a distinct clade from IA-1 and IB-1. All these three clades were grouped into IA₁ based on the traditional typing as the reviewer referred to. We edited the manuscript and made the clade names clear throughout the manuscript. To further address reviewer's comment, in **Supplementary table 1**, for the ease of comparison, we added the phylotype notations by McDowell (PLoS One 7:e41480. <http://dx.doi.org/10.1371/journal.pone.0041480>) and by Lomholt (PLoS One 5:e12277. <http://dx.doi.org/10.1371/journal.pone.0012277>), which were based on MLST analyses in previous *C. acnes* phylogenetic analysis studies. Please see modified text on **Page 4, lines 73-85**.

2. The degradation activity of HylA and HylB differed by amino acid substitutions and amino acid sequences of HylA and HylB differed by strains and its activity altered. Is there possible for that mutated HylA or HylB show degradation activity of like HylA or HylB, respectively.

3. Extended data Fig. 10. Do same genotype strains have same HylA and HylB with mutations? If an enzyme activity is dependent for pathogenicity of acne, strain's genotypes and enzyme activity may roughly match.

Response: Thanks for the query. Based on structure analysis of HylA and HylB and published literature, we arrived at a list of amino acid residue differences within the catalytic cleft that most likely contributed to difference in enzymatic activities. In Extended data Fig. 11, single amino acid substitutions were made in HylA and equivalent HylB residues, but did not identify critical residues that could explain HylA/HylB activity. More careful HPLC analyses of HA degradation products by few mutants identified residue S452 as potentially important (Figure 4): notably, mutagenesis from S to G at residue 452 (S452G) significantly reduced large HA degradation products, although this substitution did not fully convert HylA to HylB activity, since residual HA-4 remained compared to complete degradation to HA-2 by HylB.

To verify whether the role S452 in HylA is restricted to particular strains or applicable to other strains, we performed the amino acid multiple sequence alignment of the Hyl from different *C. acnes* strains (**Extended data Fig. 12**). Our analysis shows that amino acids are highly conserved across the different *C. acnes* strains. In particular, the key residues in HylA (Ser452) and HylB (Gly448) are conserved across all strains. Consistent with this observation, a report (Nazipi Microorganism 2017) have shown HylA and HylB activity from several acne- and health- associated strains that are consistent with our finding. Based on these, we believe that degradation or hydrolytic activity of mutated HylA (S452G) is also conserved across *C. acnes* strains. These results are added in the text (**Page 10, lines 214-216**).

4. Fig. 1a.

Was the strains and genetic data used from previous study? Please add appropriately reference and show the detailed data (no. of strains, strain name...).

Response: Yes, the strains, their genome data and their acne/health association were published in previous studies (Fitz-Gibbon, 2013 and Tomida, 2013). We added the references in the revised manuscript (**Page 4, line 77 and Fig. 1a legend**).

Reviewer #2 (Remarks to the Author):

The authors convincingly demonstrate the ability of HylA enzyme to cause acne when expressed by *C. acnes* bacteria, in contrast to HylB enzyme. They further clearly show that this is because HylB efficiently digests hyaluronan into disaccharides whereas HylA produces a mix of larger oligomers. As such, the degradation products of HylA activate TLR-dependent inflammatory processes, in contrast to the non-inflammatory disaccharides produced by HylB. The manuscript is additionally strengthened by the x-ray crystal structures of the two enzymes, and the discovery of peptide inhibitors of HylA. All of these factors contribute to my general enthusiasm about the manuscript. I do feel the manuscript could be strengthened by addressing the following comments:

1. What proportion of acne cases is estimated to be caused by HylA producing *C. acnes*?

Response: We were unable to find estimates of acne caused by HylA producing *C. acnes*. Our manuscript is the first to present data correlating HylA, acne-associated *C. acnes* and immunopathology. Hence, this likely explains why this correlation has not been reported.

a. The authors' efforts to derive enzymatic mechanisms from their crystal structures are not convincing, and statements that should more properly be put forth as hypotheses are instead put forth as statements of fact.

Response: We regret the oversight that it was not clearly stated as a hypothesis. In the revised version, to support the mechanistic aspect of the HylA and HylB, we have performed molecular dynamics simulations of the apo HylA-WT and HylB-WT crystal structures, and the HylA-S452G and HylA-E346G mutants. Briefly, PCA analyses of simulated trajectories suggest that HylB-WT is far more dynamic than HylA-WT and S452G mutation in HylA shows increased (amplitude) domain motions similar to HylB-WT; the cleft opening/closing motion (Eigenvector 1) increased by 20%, while the other domain motions increased by 10-40% (**Extended data Fig. 13**). These observations are consistent with the hypothesis that complex structural dynamics are one of the key mechanisms for substrate processing by HylA and HylB (Main text modified on **Pages 10 & 11, lines 217-227**; The relevant method is included on **Pages 23 & 24, lines 490-502**).

b. Based on the above logic, mutations that introduce glycine impart flexibility, which in turn enables "accelerated HA degradation." It is therefore surprising to me that, "E346G decreased the enzymatic activity of HylA." Does this finding contradict the above logic?

Response: We respectfully disagree that there is a contradiction in our logical explanation. The residue E346 is located on the inner surface of the substrate-binding cleft away from the catalytic site (N226-Y285-H276) and is likely to be involved in substrate binding in the cleft and its enzymatic activity. On the contrary, S452 is located at the site of hydrolysis. Thus, flexibility due

to the substitution of serine for glycine at this site is likely to affect the disposition of loop L4 locally and hence the efficacy of substrate binding and its enzymatic activity.

c. “the major differences between HylA and HylB, which include 346E/342G, 394A/390S, 395S/391T, 442N/438D, and 452S/448G (Extended data Fig. 11). These differences can critically alter the HA degradation mechanism, as the residues involving in the domain movements and structural flexibility can regulate the substrate entry and translocation/sliding between the subsequent catalytic cycles of the processive degradation of the polymeric/oligomeric HA substrate.” This conclusion is based upon an extrapolation from another group’s work on a homologous protein. Therefore, the tone of certainty of this conclusion does not seem justified.

Response: We thank the reviewer for pointing out the lack of data support. Since our crystal structure was determined without substrate-bound and the conclusion was based on the published work, we agree with the reviewer it may not be appropriate and justified. In the revised version the discussions related to this aspect were removed.

d. “The observation that HylA cleft is more open than the HylB cleft (Supplementary Tables 6 and 7) is in correlation with the above inference of mechanistic differences in their HA-degradation.” It is not clear to me what the author’s justification for this statement is. That is, I did not see them present a convincing argument that the difference in openness of the cleft is a meaningful correlation with the assumed mechanistic differences.

Response: We have clarified the proposed mechanism as a hypothesis and included data on the domain motions of HylA, HylB and the mutants of HylA. Please refer to the response to the critique 2a (**Pages 10 & 11, lines 217-227**).

e. “For the processive nature of degradation, HA binding must be well balanced to be strong enough to allow for HA to remain in the cleft after catalysis and weak enough to allow for sliding of the HA in the cleft for the next round of the reaction. However, the dominant electropositive charges of the HylA cleft result in very strong binding of HA that makes the sliding/translocation of the HA very difficult between the subsequent catalytic events of processive exolytic degradation. While the balanced positive and negative electrostatic charges in the HylB cleft allow the proper binding and sliding of the polymeric/oligomeric HA substrate between the subsequent catalytic events.” It seems to me that the authors do not provide any of their own data to back these statements. If this is indeed the case, I would urge them to rephrase this statement to emphasize that this is a hypothesis based on structural differences between their own apo crystal structures, and, perhaps, based on others’ work on homologs. **If the homolog work is indeed relevant, it would be useful to cite relevant findings from that work directly adjacent to this statement.**

Response: The statement has been revised as a hypothesis. We understand that the homolog work may be not relevant in this case because of the low-sequence homology between *P. acne* hylases and its homologs. Hence the related discussions were removed in the revised version.

f. Are there data supporting differences in domain motion amplitudes between HylA and HylB, or is this a hypothesis that would need to be tested?

Response: In the revised version, we tested the hypothesis using a molecular dynamics study following the procedure described by Joshi HV et al. (26). Briefly, analyses of simulation studies show that the domain motion amplitudes are higher in HylB in comparison to the respective domain motions in HylA (**Extended data Fig. 13**). The relevant method is included on **Pages 23 & 24, lines 490-502**.

g. “The i932 peptide docked in the HylA active site cleft.” Please add relevant information on docking to the Methods.

Response: We regret the omission of the method. Now, we have added the description in the main and method sections (**Page 24, lines 504-520**).

h. “These structural regions in the homologous enzymes, including ScHyl (PDB: 2X03), SpnHyl (PDB: 2BRW) and SaHyl (PDB: 1F1S), were shown to be involved in substrate attraction, binding, positioning and translocation, and product release.” If allowable under journal guidelines, please cite relevant references for this statement that is part of the caption for Extended data Fig. 9.

Response: References have been added as suggested (**Page 59, lines 984-986**).

Reviewer #3 (Remarks to the Author):

This is a very well-written paper from a leading group in *C. acnes* research. The link of hyaluronidase variants exclusively expressed by *C. acnes* strains, demonstrate remarkable clinical correlation with acne. This is novel and exciting, and the data are convincing. This is an important contribution to our field and provides further support for targeting HylA as an approach for acne therapy. However, although the topic is of general interest for the readers of the Nature Communications, the in vivo animal results are preliminary and over-interpreted. The premise for the immunization experiments is unclear as currently, there is no single animal model that encompasses all the pathogenic factors and their interactions in acne formation, which is a genuine human disease. The overarching question is whether the immunization experiments in mice are physiologically relevant in human acne? In addition: i) The immune correlates of protection in the HylA immunization experiments are not defined—which begs the question as to the physiological relevance of these experiments and if acne is a systemic disease; ii) The antibody titers and the classes of antibodies induced by the HylA injections are also not defined iii) The T cell responses induced after injection and their role in protection is not mentioned or discussed iv) There is no rationale for the design and use of the HylA multi-epitope vaccine in the experiments; why only B cell epitopes without consideration for T cell epitopes as well and iv) It is unclear if the selected B epitopes would induce cross reactive antibodies with native human proteins. Overall, the interpretation of data especially the HylA immunization experiments should be revised accordingly as they distract from an otherwise well-designed study.

Response: We agree with the reviewer that there is no single model that fully mimic human acne, although for the study of the role of *C. acnes* in human acne disease, our model is the only model that has been validated to demonstrate greater immunopathology with acne-associated strains compared to health-associated strains (Kolar et al. *JCI Insight* 2019). Since there is no model that fully mimic human disease, we have added a cautionary note that our vaccine findings need to be interpreted with caution in relation to translation (**Page 15, lines 322-324**).

Regarding the anti-HylA vaccine response, we have performed new adoptive transfer experiments to show that CD3⁺T cells from vaccinated mice are not protective (Extended data Fig. g-k). We have also provided titers of anti-HylA IgG (Fig. 6f and Extended data Fig. b and e). We have provided the rationale for design of the multi-epitope protein vaccination – selecting B cell epitopes of HylA protein that do not cross react with HylB. Based on a BLAST search, the protein does not have homology to human proteins. Please see **Page 13, lines 264-270**.

Additional comments to the authors

1. It is not clear when the cytokine levels were measured post i.d immunizations?

Response: Cytokines were measured at either 24 hr or 48 hr post-infection. This is now annotated in the figure legends and in the materials and methods section.

2. In Fig. 5, important controls are missing for example i) non-stimulated controls are lacking ii) Cytokine levels in animals injected with sebum vs. sebum + *C. acnes* HylA and HylB and/or HaCaT cells stimulated with Sebum alone

Response: Thanks. Initially, we omitted the medium controls in some of our figures as responses were similar to the HA control alone. We now provide the missing medium controls in our figures. (**Fig. 5a-c**). We also repeated an experiment comparing WT and TLR2^{-/-} BMDMs and provided the missing media control in that figure (**Fig. 5h**).

Since sebum is lipid-soluble, the effect of sebum cannot be assessed in vitro. Hence, we measured the effect of sebum alone versus sebum plus bacteria in vivo, and measured pro-inflammatory cytokines and disease score as readout (**Extended data Fig. 2c-f**).

3. Other cytokines such as IL-8 and TNF- α levels should be measured as well

Response: IL-8 (human only) and TNF- α levels are now shown (**Fig. 1f and 1j, Fig. 5b-c, and Extended data Fig. 14c and Fig. 17a, f, k**).

4. The introduction is not comprehensive and key papers in the field are not cited. the following should be cited (Page 3-line 50-51 work by Yu et al PMID: 27377696, Agak et al PMID: 28864077); Pg 3 line 64 (Kim J et al 2002, PMID 12133981); Pg 4 line 76 (Yu et al. PMID: 27377696)

Response: Thanks. In the modified introduction, we referred to work by Yu and Agak as we additionally discussed pro-inflammatory differences between health and acne-associated strains during engagement with host immune cells (**Page 3, lines 51-54**). We also referenced Kim et al. on page 3.

5. There is need to show proof that HylB is anti-inflammatory: the authors need to measure the levels of anti-inflammatory cytokines e.g., IL-10 (Pg 5 line 99-100)

Response: Thanks. We have previously demonstrated that HA-2 producing Hyl are anti-inflammatory (Kolar *Cell Host Microbe* 2015) from two mechanisms: 1) As diHA is non-inflammatory, and degradation of pro-inflammatory Hyl fragments to diHA thereby reduces inflammation; 2) diHA competes with larger-sized HA for binding to TLR2 and therefore reduces TLR2 signaling. Please see **Page 11, lines 238-240**. In a separate experiment, we have shown that IL10 induction does not account for HylB anti-inflammatory activity (**Extended data Fig. 2b**).

6. Page 5 line 106 The 2013 publication is not recent. Please edit accordingly. Page 11-line 225 fix typos

Response: We did not find a 2013 reference on line 106 in the submitted manuscript (version downloaded from the journal online site). We verified the appropriateness of the citation on page 5 line 106. The paragraph that included the typo on line 225 has been modified in response to reviewer 2 comments.

7. Page 11-line 240 key papers in the field should be cited (Kim J et al 2002, PMID 12133981)

Response: Thank you. The citation has been added.

8. The authors should make the section headings consistent

Response: We have modified the manuscript to make section headings consistent throughout.

9. Some supplemental figures need edits

Response: We have reviewed the supplemental figures and made edits, including switching the order of two figures to facilitate reading.

10. Ext. data fig. 6 Hyl expression by 43PA2 and 43PA1 is negligible via WB. Is there a better way to quantitate this?

Response: We increased the contrast to our original figure and noted in the legend that the figure is contrast enhanced for both hylA and hylB bands.

11. Ext. data fig. 7 needs caption

Response: A caption has been added to Ext. data Fig. 7.

12. Ext. data fig. 8 yellow rectangle not visible. Please select a different color

Response: We have modified the color of the rectangle in Ext. data fig. 8. to gray.

13. What is the green circle explanation in ext. Data fig. 12?

Response: Since there were no high molecular HA, we removed the green circle.

14. Table 1: Please document that some IB phlotypes express HylB

Response: Thanks for noting. IB-3 strains express HylB. Although some of these strains were associated with health, there were insufficient number to provide a firm acne or health association. IB-1 and IB-2 express HylA and are associated with acne. The requested modifications are now added to **Fig. 1a**.

15. It would up value the manuscript to include T cell data regarding the vaccine experiments.

Response: We performed adoptive transfer of CD3+T cells from mice vaccinated with protective mEHylA and showed that the CD3+T cells did not protect against HL043Pa1 induced immunopathology (**Extended data Fig. 17g-k and Page 13, lines 272-274**).

REVIEWERS' COMMENTS

Reviewer #2 (Remarks to the Author):

I thank the authors for appropriately addressing my questions/comments.

Reviewer #3 (Remarks to the Author):

Comments

Response: We agree with the reviewer that there is no single model that fully mimic human acne, although for the study of the role of *C. acnes* in human acne disease, our model is the only model that has been validated to demonstrate greater immunopathology with acne-associated strains compared to health-associated strains (Kolar et al. JCI Insight 2019). Since there is no model that fully mimic human disease, we have added a cautionary note that our vaccine findings need to be interpreted with caution in relation to translation (Page 15, lines 322-324).

A cautionary note that vaccine findings need to be interpreted with caution in relation to translation (Page 15, lines 322-324) is Acceptable.

Regarding the anti-HylA vaccine response, we have performed new adoptive transfer experiments to show that CD3+T cells from vaccinated mice are not protective (Extended data Fig. g-k). We have also provided titers of anti-HylA IgG (Fig. 6f and Extended data Fig. b and e).

The antibody titers and the subclasses of antibodies induced by the HylA injections are still not clearly defined. It unclear which Figure the authors are referring to. What is (Extended data Fig. b and e)? Are anti-HylA IgG protective against *C. acnes* infection? If yes, which subclasses are involved? Any proposed mechanism by which this antibody subclass mediate protection?

We have provided the rationale for design of the multi-epitope protein vaccination – selecting B cell epitopes of HylA protein that does not cross react with HylB. Based on a BLAST search, the protein does not have homology to human proteins. Please see Page 13, lines 264-270. This is acceptable.

Response to additional comments is acceptable.

Point-by-point response to the reviewers' comments

Reviewer #3 (Remarks to the Author):

Regarding the anti-HyIA vaccine response, we have performed new adoptive transfer experiments to show that CD3+T cells from vaccinated mice are not protective (Extended data Fig. g-k). We have also provided titers of anti-HyIA IgG.

The antibody titers and the subclasses of antibodies induced by the HyIA injections are still not clearly defined. It unclear which Figure the authors are referring to. What is (Extended data Fig. b and e)? Are anti-HyIA IgG protective against *C. acnes* infection? If yes, which subclasses are involved? Any proposed mechanism by which this antibody subclass mediate protection?

Response: We apologize for the confusion. Total IgG titers can be found in Fig. 6f (for the mEHyIA vaccine) and Suppl Fig. 15b and 15e (for the full rHyIA vaccine). In new Supplementary Fig. 18, we have added the antibody subclasses induced by mEHyIA vaccination (Suppl Fig. 18f). For mechanism, we now show that coincubation of rHyIA with mEHyIA induced serum antibodies leads to effective neutralization of hyaluronidase activity (Suppl Fig. 18a-e). The dominant IgG1 subclass that is elicited by vaccination is likely responsible for blocking hylA enzymatic activity. The new findings have been added to lines 279-282 in the result section.